# Mechanisms of ribosome stalling by SecM at multiple elongation steps

Jun Zhang[1], Xijiang Pan[1], Kaige Yan[2], Shan Sun[1], Ning Gao[2]*, Sen-Fang Sui[1]*

[1]State Key Laboratory of Membrane Biology, Center for Structural Biology, School of Life Sciences, Tsinghua University, Beijing, China; [2]Ministry of Education Key Laboratory of Protein Sciences, Center for Structural Biology, School of Life Sciences, Tsinghua University, Beijing, China

**Abstract** Regulation of translating ribosomes is a major component of gene expression control network. In *Escherichia coli*, ribosome stalling by the C-terminal arrest sequence of SecM regulates the SecA-dependent secretion pathway. Previous studies reported many residues of SecM peptide and ribosome exit tunnel are critical for stalling. However, the underlying molecular mechanism is still not clear at the atomic level. Here, we present two cryo-EM structures of the SecM-stalled ribosomes at 3.3–3.7 Å resolution, which reveal two different stalling mechanisms at distinct elongation steps of the translation cycle: one is due to the inactivation of ribosomal peptidyl-transferase center which inhibits peptide bond formation with the incoming prolyl-tRNA; the other is the prolonged residence of the peptidyl-RNA at the hybrid A/P site which inhibits the full-scale tRNA translocation. These results demonstrate an elegant control of translation cycle by regulatory peptides through a continuous, dynamic reshaping of the functional center of the ribosome.

*For correspondence: ninggao@mail.tsinghua.edu.cn (NG); suisf@mail.tsinghua.edu.cn (SFS)

**Competing interests:** The author declares that no competing interests exist

## Introduction

Translation regulation, as an essential component of the gene expression control, is usually mediated by a variety of ribosome-interacting factors and could take place at different stages during the translation cycle (*Duval et al., 2015*; *Hinnebusch and Lorsch, 2012*; *Keiler, 2015*; *Spriggs et al., 2010*; *Starosta et al., 2014*). Furthermore, protein synthesis can also be modulated by regulatory nascent peptides, as demonstrated by an increasing number of such sequences found throughout all kingdoms of life (*Cruz-Vera et al., 2011*; *Lovett and Rogers, 1996*; *Ramu et al., 2009*). One such example is a variety of ribosome arrest peptides, often encoded by upstream open reading frames (uORF) or mRNA 5′ leaders that are utilized to regulate the expression of co-transcribed genes (*Cruz-Vera et al., 2011*; *Ito and Chiba, 2013*; *Morris and Geballe, 2000*; *Wethmar, 2014*). Ribosome stalling by nascent peptides involves interactions of the regulatory peptide with the peptide exit channel on the 50S subunit, and in a few cases, requires coordination of small-molecule ligands, such as antibiotic erythromycin and tryptophan for the ErmCL and the TnaC-induced stalling, respectively (*Arenz et al., 2014a*; *Bischoff et al., 2014*; *Johansson et al., 2014*; *Seidelt et al., 2009*).

In *Escherichia coli*, SecM-mediated translation stalling is specifically employed to regulate the secretion pathway of the cell (*McNicholas et al., 1997*; *Nakatogawa and Ito, 2001*; *Oliver et al., 1998*). SecM is encoded by the 5′-end half of a bicistronic mRNA, *secM-secA*. SecA is an ATPase-dependent molecular motor helping secretory and outer membrane proteins across the cytoplasmic membrane in bacteria (*Bauer et al., 2014*; *Economou and Wickner, 1994*; *Lill et al., 1989*; *van der Wolk et al., 1997*). SecM is a secreted protein with 170 amino acids in length, including a 17-amino-acid stalling sequence [150]FSTPVWISQAQGIRAGP[166] near its C terminus, which alone is sufficient to induce stalling. The intergenic region of the mRNA between *secM* and *secA* can form a stem-loop secondary structure, which would mask the translation initiation Shine-Dalgarno (SD) sequence of

**eLife digest** Many genes code for proteins that carry out essential tasks. The instructions in a gene are first copied into a messenger RNA (mRNA), and a molecular machine known as a ribosome reads the copied instructions in groups of three letters at a time (called codons). The ribosome translates the order of the codons into a sequence of amino acids; each amino acid is carried into the ribosome by a transfer RNA (tRNA) molecule. As it translates, the ribosome joins each new amino acid to the one before it, like the links in a chain. Finally, the newly built protein chain passes through a tunnel to exit the ribosome. Ribosomes do not build all proteins at a constant rate; there are many examples of proteins that stall when they are in the ribosome exit tunnel. It is thought that this stalling is an important way for cells to control the expression of proteins.

SecM is a bacterial protein that stalls while it is being made. Previous research has shown that a sequence of amino acids in SecM (called the arrest sequence) interacts with components of the ribosome tunnel. This interaction leads to stalling, and regulates the translation of another important bacterial protein (called SecA) that is encoded downstream on the same mRNA as SecM. If SecM-induced stalling takes place, the translation of SecA actually increases. Nevertheless, it remains poorly understood how SecM stalls in the ribosome.

Zhang et al. have now solved the structures of SecM proteins stalled inside ribosomes using a method called cryo-electron microscopy. This approach identified two different states of SecM present in the ribosome, which corresponded to two different stalling mechanisms.

The addition of an amino acid to a growing protein occurs in stages. First, the tRNA that carries the amino acid to the ribosome and bind to it in a region known as the A-site. After this, the tRNA moves to the P-site where the attached amino acid is incorporated into the elongating protein chain. Zhang et al. observed that the arrest sequence of SecM and the ribosome tunnel interact extensively. These interactions are strong and alter the configuration of both the A-site and P-site of the ribosome. This has two major consequences for translation. First, the tRNA cannot be stably accommodated in the A-site and secondly, its passage to the P-site is slowed down. Both these mechanisms contribute to stalling.

This study provides a detailed analysis of how the ribosome can adjust to control translation. It also highlights that codon-specific control of translation constitutes an important component of how gene expression is regulated.

*secA* and therefore limits the synthesis of *secA* to a basal level. SecM-induced stalling could occur, as a consequence of impaired secretion pathways, and subsequently causes destabilization of the intergenic mRNA structure and increases the exposure of *secA* SD sequence to the ribosome (*Butkus et al., 2003*; *Kiser and Schmidt, 1999*; *McNicholas et al., 1997*; *Nakatogawa and Ito, 2001*, *2002*; *Sarker and Oliver, 2002*). Therefore, as a response to compromised secretion activity in the cell, the synthesis of SecA is upregulated. The stalling sequence of SecM could also induce ribosome stalling under normal conditions, but it is only temporary and quickly rescued by the functional Sec system through a simple pulling force by translocon (*Butkus et al., 2003*; *Goldman et al., 2015*). Thus, the regulatory peptide within SecM offers a feedback loop on the ribosome to ensure sufficient level of SecA in bacteria to regulate protein secretion.

Previous biochemical and structural studies have demonstrated that the ribosome stalling originates from the interaction of the 17-amino-acid nascent peptide of SecM with the 50S exit tunnel components. In the arrest sequence, R163, G165, and P166 are essential, because mutation of any of these residues can completely abolish stalling (*Nakatogawa and Ito, 2002*; *Yap and Bernstein, 2009*). Other five residues (F150, W155, I156, G161, and I162) are also important as mutations of them can abolish stalling partially (*Nakatogawa and Ito, 2002*; *Yap and Bernstein, 2009*). A few ribosomal components lining the tunnel are also required for efficient stalling, for mutations of A2058G, A2062U, or A2503G, or insertion of one adenine nucleotide within the five consecutive adenine residues (A749-A753), as well as mutations or deletion of selected residues from uL22 and uL4 could all alleviate translational stalling to certain extents (*Lawrence et al., 2008*; *Nakatogawa and Ito, 2002*; *Vazquez-Laslop et al., 2010*; *Woolhead et al., 2006*). Previous

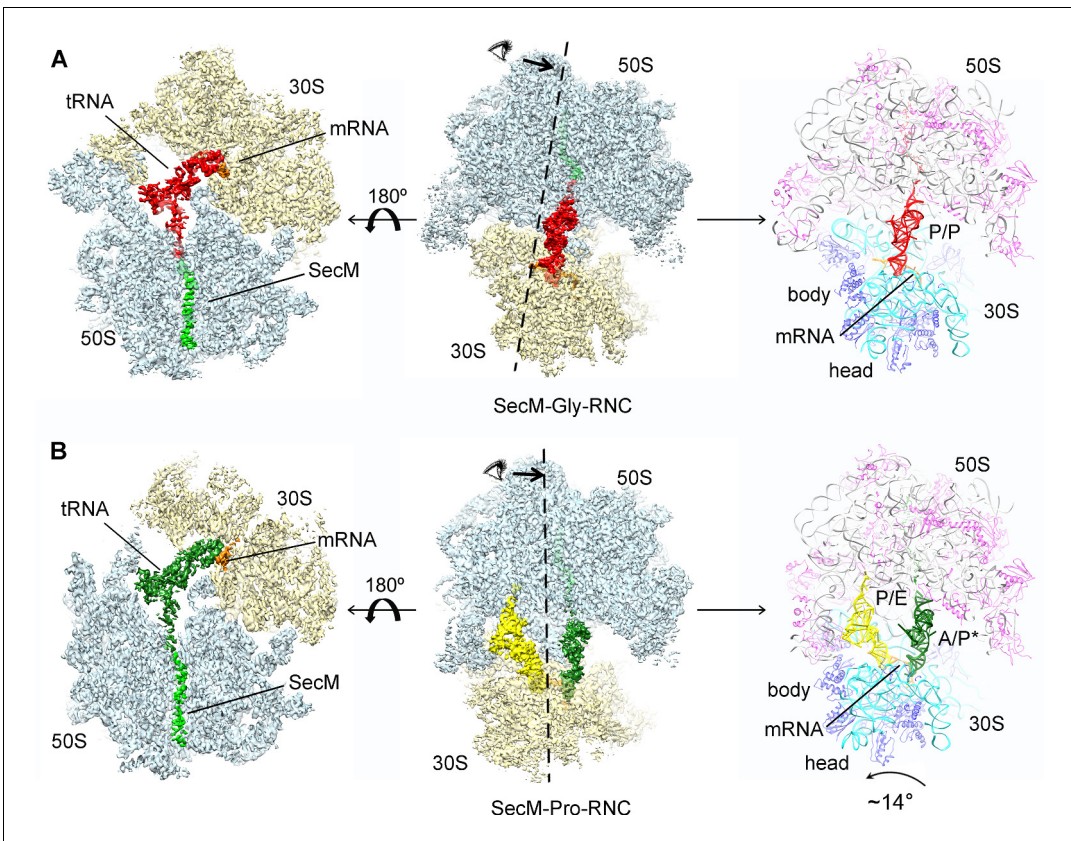

**Figure 1.** Cryo-EM structures and atomic models of SecM-stalled ribosomes. (**A**) Cryo-EM density map (3.6 Å) and atomic model of SecM-Gly-RNC, with a peptidyl-tRNA at the P/P-site. Surface representation of the map (50S, 30S, SecM-tRNA, nascent chain, and mRNA in light blue, yellow, red, green, and orange, respectively) is shown in the middle panel, and the atomic model on the right panel. The cut-away view of the density map on the left panel highlights the tunnel and the nascent peptide within. (**B**) Same as (**A**), but for SecM-Pro-RNC (3.3 Å). The A/P*-site tRNA (SecM-tRNA) and P/E-site tRNA are colored forest green and bright yellow, respectively. For atomic models, mRNA, 16S rRNA, 30S proteins, 23S rRNA, and 50S proteins are colored orange, cyan, blue, grey, and magenta, respectively.

The following figure supplements are available for figure 1:

**Figure supplement 1.** Preparation of SecM-stalled RNCs.

**Figure supplement 2.** Mass spectrometry analysis of the nascent peptides purified from SecM-stalled RNCs.

**Figure supplement 3.** Overview of the image processing.

**Figure supplement 4.** FSC curves and Cryo-EM density.

**Figure supplement 5.** Map density of the antiodon region of tRNAs and codon region of mRNA in SecM-stalled RNCs.

**Figure supplement 6.** Positions of tRNAs in SecM-stalled RNCs.

**Figure supplement 7.** Chloramphenicol is present in SecM-Pro-RNC, but not SecM-Gly-RNC.

**Figure supplement 8.** Cross-validation of the atomic model refinement.

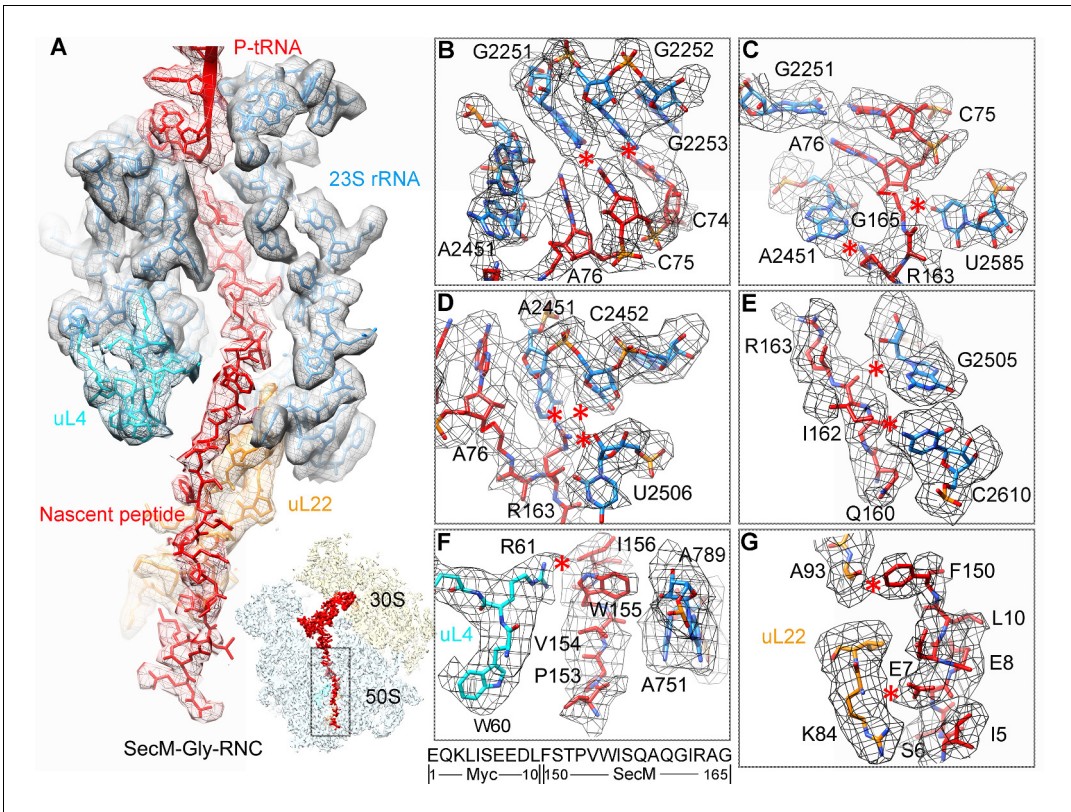

**Figure 2.** Interactions between SecM-Gly-nascent chain and ribosomal tunnel. (**A**) A zoom-in view of the density map in the regions where SecM-Gly-nascent chain interacts with exit tunnel. A transverse-section of the cryo-EM map of the SecM-Gly-RNC, showing P-tRNA and SecM-Gly-nascent chain within the ribosomal exit tunnel is provided in the lower right corner. (**B–G**) Zoom-in views of the density map in selected regions, highlighting extensive interactions between SecM and ribosomal tunnel components. Map density is shown in mesh, and the atomic model in stick representation. The coloring scheme is the same as in (**A**). Strong interactions between SecM-nascent-peptide and ribosomal components are indicated by red asterisks. The primary sequences from N-terminal Myc-tag to C-terminal of SecM peptide within the exit tunnel are shown below the panel (**F**). To illustrate the interaction between the nascent peptide and ribosomal tunnel components, the density map was displayed at relatively lower contour (1.5–3 σ).

The following figure supplement is available for figure 2:

**Figure supplement 1.** Selected regions of interactions between SecM and ribosomal tunnel components in SecM-Gly-RNC.

structural studies of the SecM-arrested ribosome suggested that the interaction between the exit tunnel and the arrest peptide could change the conformation of the PTC (peptidyl-transferase center) to slow down the peptide bond formation (*Bhushan et al., 2011*; *Gumbart et al., 2012*). However, the previous structures were not in sufficient resolution for direct visualization of the atomic interactions between the tunnel components and the nascent peptide. Furthermore, a recent study employed fluorescence resonance energy transfer (FRET) to monitor the real-time translation of SecM on the ribosome (*Tsai et al., 2014*), and revealed that the stalling is a dynamic process involving reduced elongation rates at a range of positions on the SecM mRNA, from G165 to 4–5 codons after the terminal P166 of the arrest sequence, including increased lifetime for both unrotated and rotated ribosomes at these codon positions. Nevertheless, although the stalling induced by SecM is not strictly a single-site event, G165 is the first predominant site of stalling (*Tsai et al., 2014*).

Recent advancement of cryo-EM single particle technique, such as the application of direct electron detection devices and efficient algorithms for conformational sorting of particles allow simultaneous high-resolution structural determination of several functional states from a single

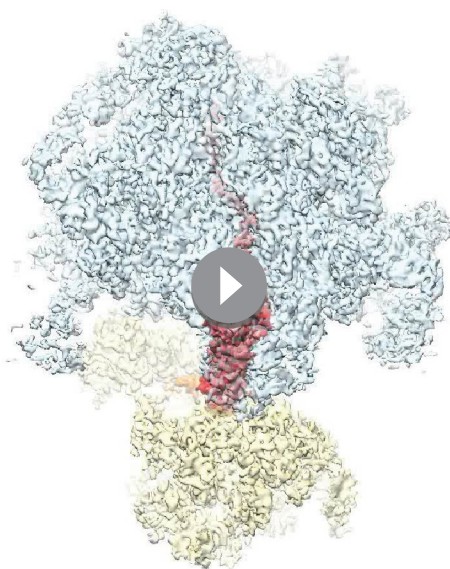

**Video 1.** Cryo-EM density map of SecM-Gly-RNC. The cryo-EM density map of SecM-Gly-RNC is shown in transparent surface representation. The 50S subunit, 30S subunit, SecM-Gly-tRNA, and mRNA are colored pale blue, yellow, red, and orange, respectively. In the final movie frames, the map is shown in a cut-away view and zoomed into the region of the peptide exit tunnel, highlighting the densities for the nascent chain.

heterogeneous dataset (*Bai et al., 2015*; *Cheng, 2015*; *Cheng et al., 2015*). Therefore, we set out to use this method to analyze the structures of the ribosomes stalled on SecM mRNA. Our structural data of the two predominant forms of stalled ribosomes, one in post-state, and the other in hybrid rotated state, indicate that a collection of interactions between SecM and the exit tunnel cooperatively induce conformational changes of the PTC, leading to translation arrest at distinct elongation steps, including peptide-bond formation and tRNA translocation.

## Results

### Biochemical sample preparation and cryo-EM structural determination

To understand the molecular mechanism of SecM-dependent translational stalling, we set out to purify SecM-stalled ribosome nascent chain complexes (RNCs) using an in vitro translation system from *E. coli* and to analyze their structures using single particle cryo-EM technique. To facilitate biochemical characterization and purification, two similar constructs were prepared: one encodes, from the N- to C-terminus, a 2xStrep-TEV-tag, the N-terminal 40 residues of OmpA, a Myc-tag, SecM stalling sequence (residues 150–166) and tandem stop codons (SEC-STOP); the other contains an additional 6X-His-tag (SEC-HIS-STOP) after SecM stalling sequence> (*Figure 1— figure supplement 1A*) . After incubation of the plasmids with the S30-T7-based in vitro translation system, the presence and the amount of arrested peptidyl-tRNA can be detected using Western blot with primary antibody against Myc-tag (*Figure 1— figure supplement 1B–E*). We found that the peptidyl-tRNA in the reaction mixture started to accumulate after 5 min, peaked at 15 min and became diminished after 30 min. Therefore, we chose 15 min as the incubation time to purify SecM-stalled RNCs. The design of the SEC-HIS-STOP construct was to assess how frequent the ribosome would go beyond P166. As shown in *Figure 1— figure supplement 1B*, lower band corresponding to SecM-HIS peptide also existed in the reaction mixtures, in a comparable amount to the peptidyl-tRNA, indicating the ribosome was able to go beyond SecM coding sequence and reach the stop codon with increasing incubation time. However, after treatment of the samples with excessive RNase A, which destroys all RNAs and releases the peptide from the peptidyl-tRNA, only the band of SecM peptide, but not SecM-HIS, sharply increased, indicating that the peptidyl-tRNA in the reaction mixtures is indeed arrested on the very 3-end of SecM coding sequence. The construct of SEC-STOP was specifically used to further synchronize the RNCs on the terminal codon to provide a homogeneous complex for structural analysis. As designed, the peptidyl-tRNA in the reaction mixture of SEC-STOP displayed exactly the same motility on the gel as the one from SEC-HIS-STOP. The RNCs were isolated by a sucrose cushion followed by Strep-affinity chromatography, and further separated and enriched by a second round of sucrose cushion (*Figure 1— figure supplement 1C,D*). Similar as previous studies of ribosome stalling (*Bhushan et al., 2011*; *Frauenfeld et al., 2011*; *Halic et al., 2006*; *Yap and Bernstein, 2009*), chloramphenicol was added in the sucrose cushion buffer to stabilize the RNC and to minimize the peptidyl-tRNA hydrolysis during the lengthy purification. As expected, Western blotting indicates that a clean single band of the peptidyl-tRNA was detected in the purified RNCs (*Figure 1— figure supplement 1D*), indicating the compositional homogeneity of the purified RNC. To identify nascent peptides in the purified

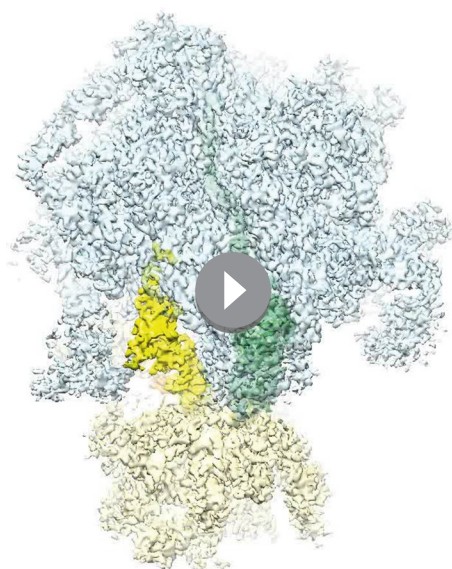

**Video 2.** Cryo-EM density map of SecM-Pro-RNC. The cryo-EM density map of SecM-Pro-RNC is shown in transparent surface representation. The 50S subunit, 30S subunit, SecM-Pro-tRNA (A/P*), P/E-tRNA, and mRNA are colored pale blue, yellow, green, bright yellow, and orange, respectively. In the final movie frames, the map is shown in a cut-away view and zoomed into the region of the peptide exit tunnel, highlighting the densities for the nascent chain.

RNCs, we further purified nascent peptides from SecM-stalled RNCs (*Figure 1— figure supplement 2A,B*) and subjected them to mass spectrometry analysis. The C-terminal sequencing of nascent peptides using a tandem MS/MS (*Figure 1— figure supplement 2C–F*) revealed two species, with their C-termini ending at G165 and P166, respectively.

Next, we applied cryo-EM to analyze purified SecM-RNCs. A cryo-EM data set, composed of 238,212 particles, were subjected to maximum-likelihood-based 2D and 3D sorting (*Scheres, 2010*, *2012*), and as a result (*Figure 1— figure supplement 3*), two cryo-EM density maps, corresponding to the two major populations of the data, one with the ribosome in classical unrotated state (a peptidyl-tRNA at the P-site), the other with the ribosome in rotated state (two hybrid tRNAs), were obtained at resolution of 3.7 and 3.3 Å, respectively (*Figure 1A,B* and *Figure 1— figure supplement 4A–D*). These two maps have well-resolved densities for the ribosome and tRNA, with a clear separation of most of the nucleotide bases and protein sidechains (*Figure 1— figure supplement 4E–H*). To further improve the density quality of the nascent chains, a soft mask of the 50S subunit (plus tRNA) was applied during refinement (*Supplementary file 1*), similarly as previously described (*Amunts et al., 2014*;

*Brown et al., 2014*; *Greber et al., 2014a*; *Greber et al., 2014b*; *Voorhees et al., 2014*), which resulted in slightly improved overall resolutions (*Figure 1— figure supplement 4A,D*), but considerably enhanced densities for the nascent chains. The improved structures allowed us to trace and de novo model the nascent chains within the tunnel (*Video 1–3*) (see 'Materials and methods for model validation details).

Based on the information from mass spectrometry and structural modeling of the two density maps, the two structures were termed as SecM-Gly-RNC and SecM-Pro-RNC according to the identity of the peptidyl-tRNA (Gly or Pro) in the RNCs (*Figure 1—figure supplement 3*), following the same notion in the previous work (*Bhushan et al., 2011*). This assignment of the two maps was supported by two sets of structural observations. First, the local densities of the mRNA: tRNA duplex at the PTC in the two maps could be best explained by our assignment, because the anticodon of tRNA$^{pro}$ (GGA) is composed of purines exclusively, in contrast to that of tRNA$^{gly}$ (CCG). In our map of SecM-Pro-RNC, bases of the three A-site anti-codon residues indeed appears to be relatively larger (*Figure 1—figure supplement 5F*), compared with those in SecM-Gly-RNC (*Figure 1—figure supplement 5D*). Vice versa, analysis of local density in the A- and P-site codons of the mRNAs in the two maps also supports our assignment (*Figure 1—figure supplement 5A–C*). Second, two bulky residues (W155 and F150) within SecM arrest peptide could be readily located in the exit channel of the two density maps, and their positions are incompatible with other possible assignments. The peptidyl-tRNA in SecM-Gly-RNC is at the classical P site (P/P state) (*Figure 1—figure supplement 6A,B*), while the two tRNA in SecM-Pro-RNC are at hybrid states (*Figure 1—figure supplement 6C–G*). In SecM-Pro-RNC, the peptidyl-tRNA is at a position similar to previously reported A/P* hybrid state (*Brilot et al., 2013*) (*Figure 1—figure supplement 6F,G*), and the other tRNA is at a position between P/E and pe/E hybrid states (*Polikanov et al., 2014*) (*Figure 1—figure supplement 6E*), closer to the P/E configuration. In the dataset, the hybrid state of SecM-Pro-RNC accounts for 36% of total particles, with a ratio of 1:1.5 for SecM-Pro-RNC (rotated): SecM-Gly-RNC (unrotated). Also comparable to the previous observation (*Bhushan et al., 2011*), a minor class of particles

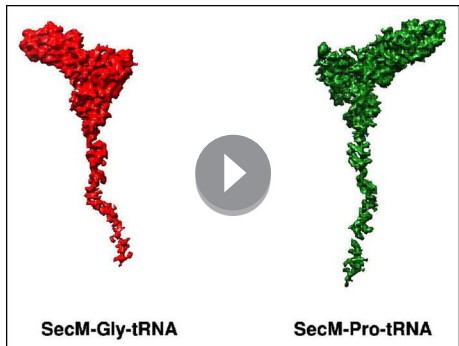

**Video 3.** Atomic models of the nascent peptides in SecM-Gly-RNC and SecM-Pro-RNC. Segmented density maps of the peptidyl-tRNAs in SecM-Gly-RNC (red) and in SecM-Pro-RNC (green) are shown in surface representation, with C-terminal half of SecM peptide highlighted in zoom-in views. Atomic models of the nascent peptides are superimposed with the density maps, with a few landmark residues of SecM labeled.

resulted in a low-resolution structure of a pre-state ribosome (unrotated with A/A and P/P-tRNAs), likely representing SecM-Gly-RNC with prolyl-tRNA at the A-site (*Figure 1—figure supplement 3*). It should be noted that during a normal cycle of elongation, the fraction of rotated ribosomes is much less (*Fischer et al., 2010*; *Kim et al., 2007*; *Munro et al., 2007*), as it reflects a high-energy state and requires stabilization from translation factor (*Frank et al., 2007*). Therefore, the increased lifetime of the A/P*, P/E intermediate states of tRNAs in SecM-Pro-RNC suggests this structure might represent a stalled form that is trapped in an intermediate translocational state.

## Interactions of SecM with the peptide exit tunnel in SecM-Gly-RNC

In SecM-Gly-RNC, extensive interactions between the nascent chain and the tunnel could be readily identified (*Figure 2A*). The first region involves the PTC, the CCA end of the peptidyl-tRNA and R163 of SecM. The CCA-end of the P-tRNA forms canonical base pairing with the P-loop (*Figure 2B*); the base of U2585 contacts the peptidyl bond linkage between G165 and A76 of the P-site tRNA (*Figure 2C*); R163 is situated at a space confined by three nucleotides, U2506, A2451, and C2452 (*Figure 2D*). These extensive interactions involving R163-G165 perfectly explain their critical roles in SecM-induced stalling (*Nakatogawa and Ito, 2002*; *Yap and Bernstein, 2009*). Immediately below R163 along the tunnel, strong density connections can be observed between the carbonyl oxygen of I162 and the sugar ring of G2505, and between the sidechain of I162 and the base of C2610 (*Figure 2E*). Further downstream, multiple interactions between Q160-Q158 and U2609, C2610, C2611, and A2062 could be identified (*Figure 2E* and *Figure 2—figure supplement 1A,B*), consistent with previous mutational data that perturbation of this interface either at the tunnel (A2062U) or the nascent peptide (Q160P) impairs translation stalling (*Vazquez-Laslop et al., 2010*; *Woolhead et al., 2006*; *Yap and Bernstein, 2009*). In the mid-tunnel region, the sidechain of W155 interacts with the side-chain of R61 of uL4 (*Figure 2F*), and the backbone of I156-W155-S154 is also close to several tunnel components, including A751 and G91-R92 of uL22 (*Figure 2F* and *Figure 2—figure supplement 1C*). These observations could well explain the previous finding that mutations of either W155 or I156 into alanine can rescue the SecM-induced stalling (*Nakatogawa and Ito, 2002*). In the lower region of the tunnel, a strong hydrophobic contact between F150 and A93 of uL22 is evident (*Figure 2G*). Again, the contribution of uL22 residues (G91 and A93) in this region to the stalling was previously demonstrated by mutational data (*Lawrence et al., 2008*). Lastly, beyond F150, the last engineered residue of SecM, two sets of contacts between the Myc-tag and the tunnel could also be observed. One involves K84-I85 of uL22 and E7-I5 of the Myc sequence (*Figure 2G*), and the other one involves E1-K3 of the Myc sequence, A508 and A1321 of the 23S rRNA (*Figure 2—figure supplement 1D*). The N-terminal sequences of SecM beyond F150 are poorly conserved and not essential for stalling. However, previous data showed that SecM$_{150-166}$ is less efficient in stalling than SecM$_{140-166}$ (*Nakatogawa and Ito, 2002*), indicating that the interactions beyond F150 as we observed here, although might not be sequence-specific, also contributes to the stalling.

## Interactions of SecM with the peptide exit tunnel in SecM-Pro-RNC

Most of the residues of SecM contributing to interact with the tunnel in SecM-Gly-RNC also extensively interact with the tunnel in SecM-Pro-RNC, but with completely different patterns (*Figure 3A*). At the PTC, the CCA-end of the A/P*-tRNA is not orientated for canonical base pairing between C74-C75 of the peptidyl-tRNA and G2252-G2251 of the P-loop (*Figure 3B*). A strong density

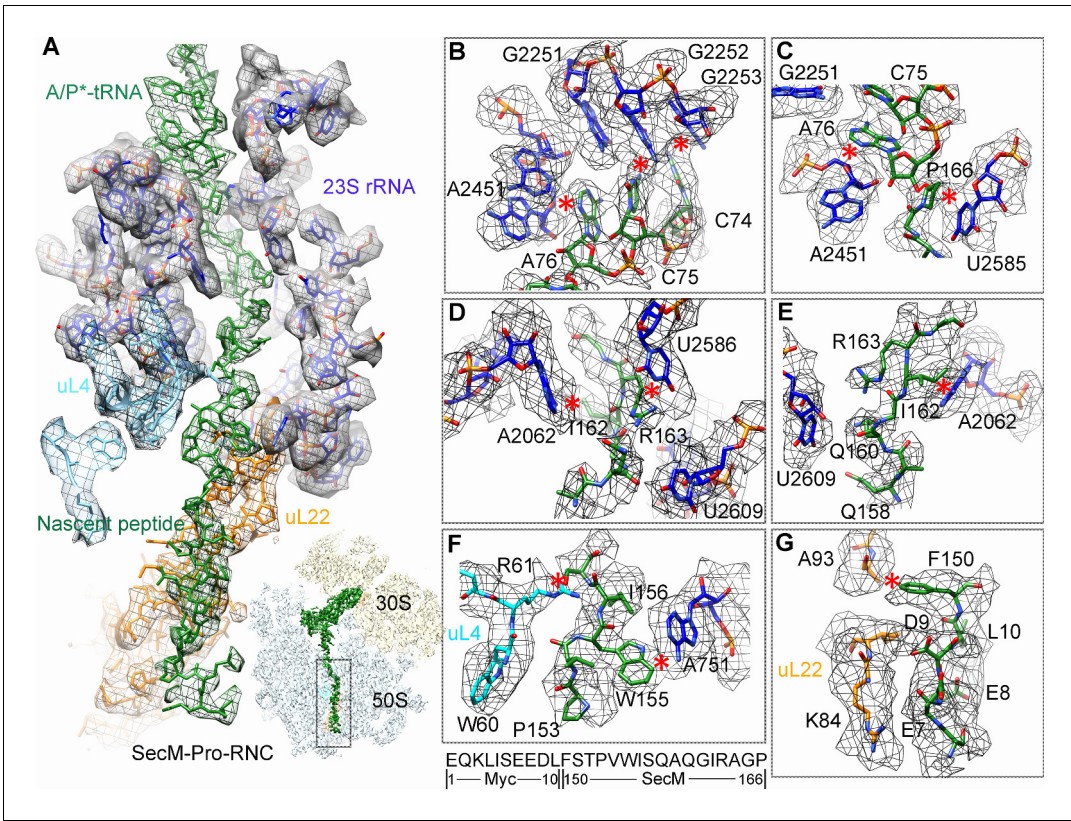

**Figure 3.** Interactions between SecM-Pro-nascent chain with ribosomal tunnel. (**A**) A zoom-in view of the density map in the region where SecM-Pro-nascent chain interacts with ribosomal tunnel. A transverse-section of the cryo-EM map of the SecM-Pro-RNC, showing P-tRNA and SecM-Pro-nascent chain within the ribosomal exit tunnel is provided in the lower right panel. (**B–G**) Zoom-in views of the density map in selected regions, highlighting extensive interactions between SecM and ribosomal tunnel components. Map density is shown in mesh, and the atomic model in stick representation. The coloring scheme is the same as in (**A**). Strong interactions between SecM-nascent-peptide and ribosomal components are indicated by red asterisks. The primary sequences from N-terminal Myc-tag to C-terminal of SecM peptide within the exit tunnel are shown below the panel (**F**). To illustrate the interaction between the nascent peptide and ribosomal tunnel components, the density map was displayed at relatively lower contour (1.5–3 σ).

The following figure supplement is available for figure 3:

**Figure supplement 1.** Selected regions of interactions between SecM and ribosomal tunnel components in SecM-Pro-RNC.

connection is seen between the base of U2585 and P166 (*Figure 3C*), which might be responsible for the distorted conformation of the CCA-end of the A/P*-tRNA. Compared with SecM-Gly-RNC, with addition of one more amino acid, G165 moves into a pocket formed by the bases of U2506 and A2451 (*Figure 3—figure supplement 1A*), while R163 relocates its sidechain between the bases of U2586 and U2609 (*Figure 3D*). In addition, strong density connections can be observed between the sidechain of I162 and the base of A2062 (*Figure 3D,E*). Further downstream, Q160 and Q158, through their sidechains, form hydrogen bonds with the bases of U2609 and A752, respectively (*Figure 3—figure supplement 1B,C*). Within the mid-tunnel region, the sidechain of R61 of uL4, instead of interacting with W155 in SecM-Gly-RNC, is at a distance capable of hydrogen bonding with the carbonyl oxygen of I156 (*Figure 2F* and *Figure 3—figure supplement 1D*). The sidechain of W155, in contrast to its location in SecM-Gly-RNC, flips ~90° and stacks with the base of A751 (*Figure 3F* and *Figure 3—figure supplement 1D*). The pattern of interactions at F150 and beyond is very similar with slight variation (*Figure 3G* and *Figure 3—figure supplement 1E*). F150 still maintains hydrophobic interaction with A93 of uL22 (*Figure 3G*), but the distance is significantly larger than that in

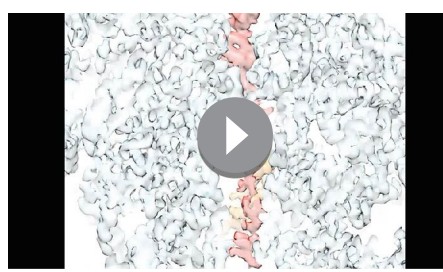

**Video 4.** Comparison of the PTC in SecM-Gly-RNC with that of the induced state. Superimposition of the atomic model of SecM-Gly-RNC with that of the induced state (1VQN) (*Schmeing et al., 2005b*) in the PTC region. Morphing between two models was shown. Critical residues of SecM and the 23S rRNA are labeled. Also see *Figure 7* and text for details.

SecM-Gly-RNC. The C- and N-terminal sequences of the Myc-tag interact similarly with K84-I85 of uL22 and A1321 of the 23S rRNA, respectively (*Figure 3G* and *Figure 3—figure supplement 1E*).

The atomic details in SecM-Pro-RNC are very different from those in SecM-Gly-RNC, especially for the C-terminal half of the arrest sequence. Actually, many of contacts observed in SecM-Pro-RNC have been previously predicted by a molecular dynamics simulation study (*Gumbart et al., 2012*), indicating that SecM-Pro-RNC is likely a thermodynamically favored state. Through analysis of the two structures, there appears to be two common constrict sites on the tunnel wall, and the interactions at these locations in the two structures are highly residue-specific: one is formed by A751 of 23S rRNA and R61 of uL4, and the other by A93 and

K84 of uL22. The presence of these specific constrict interactions in the two structures suggests that the bulky residues of SecM, W155, and F150 in particular, function to contain the peptide to slow down its passage within the tunnel to posit N-terminal R163, G165, and P166 at specific locations to induce stalling.

## Inactivation of the PTC in SecM-Gly-RNC

Peptide bond formation in the ribosome requires precise positioning of the peptidyl-tRNA and the A-tRNA (*Polikanov et al., 2014*; *Schmeing et al., 2005a*; *Schmeing et al., 2005b*). Highly conserved rRNA residues in the PTC adopt specific conformations in different functional states to determine the catalytic kinetics of the peptide bond formation. To understand how the interactions between SecM and ribosomal components observed in our models contribute to the translation stalling, we analyzed conformational differences of the PTC in SecM-Gly-RNC and SecM-Pro-RNC from the previously characterized induced (PDB ID 1VQN) and uninduced states of PTC (PDB ID 1VQ6) (*Figure 4A,C*) (*Schmeing et al., 2005b*). These two states, obtained by structural determination of the active 50S subunit (*Schmeing et al., 2005a*; *Schmeing et al., 2005b*) or the ribosome (*Polikanov et al., 2014*) with different A-site and P-site tRNAs or analogues, reflect two major states of the PTC, inactive and active for peptide bond formation. When the PTC transits from the uninduced state to induced state, U2585 and U2584 would shift away by 3–4 Å to allow the A-tRNA accommodation (*Figure 4A*), and U2506 would rotate toward the P-tRNA (*Schmeing et al., 2005b*; *Voorhees et al., 2009*) (*Figure 4C*). By comparing the conformation of PTC in SecM-Gly-RNC with those of the uninduced and induced states, we found that both U2585 (*Figure 4B*) and U2506 (*Figure 4D*) in SecM-Gly-RNC adopt positions similar as in the uninduced state (*Video 4*). Therefore, these interactions originated from G165-R163 are likely responsible for constricting the PTC of SecM-Gly-RNC, especially U2585 and U2506, into an uninduced state that disfavors A-tRNA accommodation and subsequent peptide bond formation. Moreover, the extent of U2585 shifting is even more toward the inactive form of the uninduced state, with a 70° flip compared with that of the induced state (*Figure 4—figure supplement 1A*). Similar

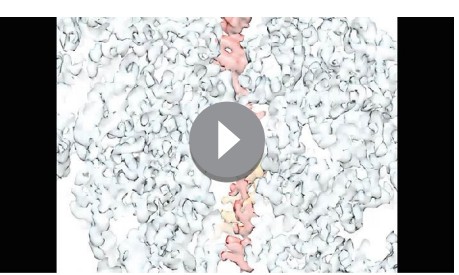

**Video 5.** Comparison of the PTC in SecM-Pro-RNC with that of the induced state. Superimposition of the atomic model of SecM-Pro-RNC with that of the induced state (1VQN) (*Schmeing et al., 2005b*) in the PTC region. Morphing between two models was shown. Critical residues of SecM and the 23S rRNA are labeled. As shown, a deformation of the CCA-end of the peptidyl-tRNA disrupts canonical P-loop interaction. Also see *Figure 7* and text for details.

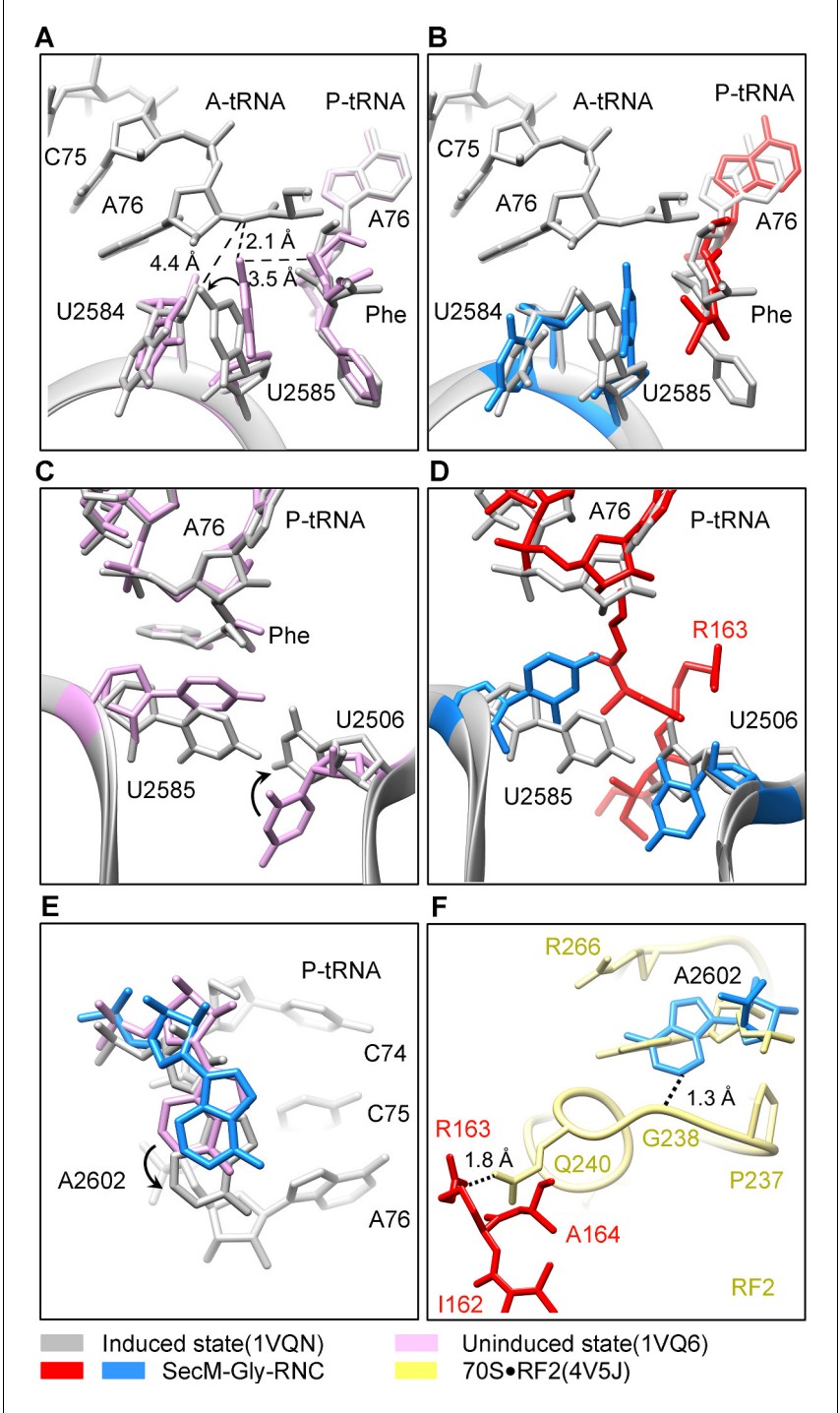

**Figure 4.** SecM stabilizes the PTC of SecM-Gly-RNC in an inactive state. (**A**) Conformational difference between the uninduced (PDB ID 1VQ6, plum) (*Schmeing et al., 2005a*) and the induced states (PDB ID 1VQN, gray) (*Schmeing et al., 2005a*) of the PTC. Direction of the shift for U2585 between two states and corresponding distances are labeled. As shown in (**A**), U2585 and U2584 would shift away by 3–4 Å from the uninduced state to the induced state. (**B**) U2584 and U2585 in SecM-Gly-RNC (23S rRNA in blue and peptidyl-tRNA in red) shift away from their regular positions in the induced state (grey) and assume an uninduced state. (**C**) Rotation of U2506 accompanying the shift of U2585 as in (**A**), upon transition from the uninduced (plum) to induced (grey) states. (**D**) Position and interaction of R163 in SecM-Gly-RNC. R163 leads to a rotation of U2506 to assume an uninduced conformation. (**E**) Comparison of A2602 in SecM-Gly-RNC with that of the uninduced and induced states. (**F**) The conformation of A2602 (blue) in SecM-Gly-RNC is incompatible with the binding of release factor2 (RF2) to the

*Figure 4 continued on next page*

*Figure 4 continued*
PTC. The position of R163 (red) in SecM-Gly-RNC is in clash with the GGQ motif of RF2 (bright yellow). The coordinates of RF2 is from a crystal structure of the 70S·RF2 complex (PDB, ID 4V5J) (*Jin et al., 2010*). The alignment was done using the 23S rRNA residues 2400–2800 as reference. The RMS deviations between respective reference sequences are 0.3 Å for 1VQN vs 1VQ6, 1.5 Å for SecM-Gly-RNC vs 1VQN, 1.0 Å for SecM-Gly-RNC vs 4V5J.
The following figure supplement is available for figure 4:

**Figure supplement 1.** Conformational changes of the PTC nucleotides in SecM-stalled RNCs.

observation is also true for U2506 in SecM-Gly-RNC (*Figure 4—figure supplement 1B*).

Similar to TnaC- and MifM-stalled ribosomes (*Bischoff et al., 2014*; *Sohmen et al., 2015*), A2602 in SecM-Gly-RNC adopts a position (*Figure 4E* and *Figure 4—figure supplement 1C*), which would block the entry of the incoming aminoacyl-tRNA to the A-site of the PTC. This specific configuration of the PTC in SecM-Gly-RNC is also incompatible with the binding of release factors, as the GGQ motif of RF2 (PDB ID 4V5J) (*Korostelev et al., 2008*) sterically clashes with A2062, as well as with R163 (*Figure 4F*). These observations provide a possible explanation why SecM-stalled ribosomes are stable over an hour (*Evans et al., 2005*; *Tsai et al., 2014*; *Uemura et al., 2008*), and resistant to the hydrolysis of peptidyl-tRNA by puromycin (*Muto et al., 2006*; *Tsai et al., 2014*).

## Destabilized interaction between the A*/P-tRNA and the P-loop in SecM-Pro-RNC

The geometry of the PTC in SecM-Pro-RNC also changed, but differently from SecM-Gly-RNC. First, U2585 shifts slightly toward the direction of the induced state, but still in a state closer to the uninduced conformation (*Figure 4—figure supplement 1A* and *Figure 5—figure supplement 1B*). Local conformational difference at U2585 and U2586 is evidently due to the interactions with P166 and R163 of SecM (*Figure 5A,C*). Second, In contrast to SecM-Gly-RNC, U2506 now assumes an induced state (*Figure 4—figure supplement 1B*, *Figure 5—figure supplement 1D*). Third, the interaction between SecM-I162 and A2062 stabilizes the latter in a state different from the induced state (*Figure 5A,D*). The most intriguing observation is that the base pairing between the CCA-end of A/P*-tRNA and the P-loop is disrupted (*Figures 3A*, *5E*, *Video 5*). The base of C75 has a twist and is seen to extend toward the base of G2252 instead of G2251 (*Figure 5E*), and C74 also deviates from its optimal base-pairing position with G2252. The destabilization of the interaction between the A/P*-tRNA with the P-loop appears to result from a cascade of interactions described in the above. Specifically, R163 and P166 of SecM are likely responsible for preventing the establishment of a stable P-loop interaction in SecM-Pro-RNC.

Taken together with the kinetic data that SecM increases the lifetime of rotated state of the ribosome (*Tsai et al., 2014*), our structural data on SecM-Pro-RNC therefore suggest that SecM might also impair the hybrid peptidyl-tRNA accommodation to the P-loop to slow down the tRNA translocation, in addition to its inhibition on peptide bond formation in SecM-Gly-RNC.

## Discussion

### Conformational changes of the ribosome upon SecM recognition by the 50S tunnel

In the present work, we obtained two high-resolution structures of SecM-arrested ribosomes, one in the post-translocational state and the other in the rotated state.

The first one, SecM-Gly-RNC, contains a peptidyl-tRNA in classical P/P-site and therefore is expected to display an overall conformation similar to the classical PRE/POST states of the ribosome. However, compared with the cryo-EM solution structures of the pre-translocational ribosome (three tRNAs) (*Brilot et al., 2013*) or the 70S·EF-Tu complex (*Villa et al., 2009*) from *E. coli*, an apparent mode of motion for the 30S subunit could be identified, featuring a rotational movement of the 30S body domain along its long axis (*Figure 6A,B*). Notably, the specific conformation of the post-state ribosome was previously captured in *E. coli* crystal structures of the 70S-EF-G/GMPPCP

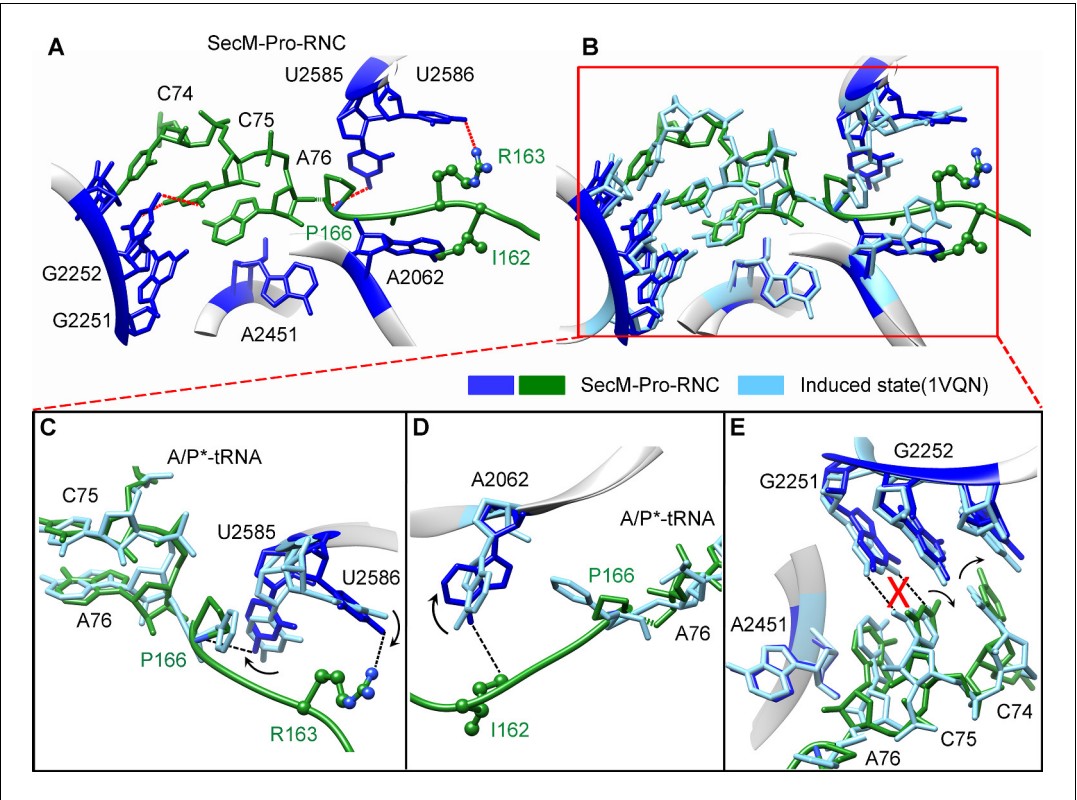

**Figure 5.** SecM destabilizes the interaction between CCA-end of the A/P*-tRNA and the P-loop in SecM-Pro-RNC. (**A**) Interactions between SecM nascent peptide and ribosomal components at the PTC in SecM-Pro-RNC. Peptidyl-tRNA and 23S rRNA are colored forest green and blue, respectively. (**B**) Comparison of the 23S rRNA nucleotides in SecM-Pro-RNC with that of the induced state of the PTC (PDB 1VQN, cyan) (*Schmeing et al., 2005a*). (**C–E**) Zoom-in views of (**B**), with orientations optimized to show the conformational difference for U2585-U2586 (**C**), A2062 (**D**), and the P-loop (**E**). The alignment was done using the 23S rRNA residues 2400–2800 as reference. The RMS deviation between respective reference sequences is 1.4 Å for SecM-Pro-RNC vs 1VQN. DOI: 10.7554/eLife.09684.022

The following figure supplement is available for figure 5:

**Figure supplement 1.** Conformation states of the PTC nucleotides in SecM-Pro-RNC around the PTC. DOI: 10.7554/eLife.09684.023

complex (*Pulk and Cate, 2013*). Moreover, compared with the *E. coli* crystal structure of the post-initiation ribosome (with one P/P-tRNA) (*Dunkle et al., 2011*), similar movement of the 30S subunit is also evident (*Figure 6C,D*). This movement could result in a displacement of the peripheral parts of the 30S subunit up to ~5 Å distance, and increase the spacing between the two subunits to a certain extent. In contrast, the central part of the 30S subunit (platform domain) remains largely unchanged (*Figure 6C–F*). Furthermore, similar rotational movement of the 30S body also prevails (*Figure 6—figure supplement 1*), when compared with *Thermus thermophilus* crystal structures of the ribosome in pre-translocational state (*Jenner et al., 2010*; *Selmer et al., 2006*), the post-translocationl state bound with EF-G (*Gao et al., 2009*), or the 70S·EF-Tu complex (*Voorhees et al., 2010*). Very interestingly, this observation is very similar to a recent finding that mammalian ribosomes possess a novel motion mode for the small subunit, named 'subunit rolling', to distinguish its POST and PRE state, and to regulate the elongation cycle (*Budkevich et al., 2014*). However, due to the lack of reference structures (especially high-resolution cryo-EM solution structures of the ribosomes in the POST and PRE states), we were unable to determine whether the rolling is a consequence of the stalling, or a general feature intrinsic to the POST-state ribosome. Nevertheless, this suggests the 'subunit-rolling' is another common mode of motion for both prokaryotic and eukaryotic ribosomes.

The other structure, SecM-Pro-RNC, contains a peptidyl-tRNA in A/P* site and a tRNA between P/E site and pe/E site, displaying a typical rotated conformation. Compared with the crystal structure

of the hybrid-state ribosome (*Dunkle et al., 2011*), or the cryo-EM structure of the pre-translocational state trapped with elongation factor-G and viomycin (*Brilot et al., 2013*), the degree of 30S subunit rotation relatively to the 50S subunit, the configuration of the intersubunit bridges, and the interactions between tRNA, 16S rRNA and mRNA in the decoding center are all highly similar (*Figure 6—figure supplement 2* and *Figure 6—figure supplement 3*).

Therefore, the similarity of the overall conformations of SecM-Gly-RNC and SecM-Pro-RNC to known structures of functional ribosomal complexes (both unrotated and rotated) suggests that SecM-induced stalling is likely due to the localized conformational changes of the PTC, induced by extensive interactions between the tunnel components and the nascent chain.

## Mechanism of SecM-induced translation stalling in SecM-Gly-RNC

Early low-resolution cryo-EM data suggested that recognition of SecM by the tunnel induces a cascade of rRNA rearrangements, propagating from the exit tunnel throughout the large subunit to reorient small subunit rRNA elements, and directly lock the ribosome and mRNAtRNAs complex (*Mitra et al., 2006*). This hypothesis was argued by improved cryo-EM structure of SecM-arrested ribosome (SecM-Gly-RNC) at 5.6 Å resolution (*Bhushan et al., 2011*), which did not reveal any significant, systematic, and large-scale rRNA arrangement on the 50S subunit. Instead, it was proposed that an observed ~2 Å shift in the position of the tRNA-nascent peptide linkage of SecM-tRNA away from the A-tRNA reduces the rate of peptide bond formation, resulting in translation stalling.

However, with much improved resolution, we did not observe significant movement of the CCA-end of the P-site tRNA. Instead, the position of the tRNA-nascent peptide linkage of SecM-tRNA is similar to that seen in the structure of the ribosome with active PTC (*Figure 4B*) (*Polikanov et al., 2014*). Through the comparison with reference states of the ribosomes that are capable or incapable of peptide-bond formation, we could provide an alternative mechanism of SecM-induced stalling in SecM-Gly-RNC (*Figure 7A*).

When the ribosome encounters the SecM arrest sequence, W155 and F150 are fixed at relatively narrow constrict sites in the middle tunnel region, which in turn causes the compaction of the C-terminal half of the peptide (*Woolhead et al., 2006*). As a result, R163 is confined in a specific position surrounded by U2506/C2452/A2451 (*Figure 2D*), leading to a strong interaction between U2585 and the O3 in A76 of CCA-end of the P-site tRNA. These altogether inactivate the PTC in an uninduced state. Particularly, positions of the U2585/U2506 and R163 would create a steric hindrance for the precise accommodation of the incoming aminoacyl-tRNA to the A-site.

In addition, when an aminoacyl-tRNA, which should be proline-tRNA in the case of SecM, reaches the A-site of the PTC, the naturally slow peptide bond formation rate for proline (*Pavlov et al., 2009*) would further enhance the stalling. Indeed, the intersubunit FRET experiments showed that the lifetime of the unrotated ribosome would significantly increase when the codon of P166 is positioned at the A-site (*Tsai et al., 2014*). Therefore, the inhibitive conformation of the PTC and low peptide bond formation rate of P166 collectively causes the stalling of the translation in SecM-Gly-RNC.

As noted in previous cryo-EM study (*Bhushan et al., 2011*), only a small fraction of ribosomes contains both A/A and P/P tRNAs in the sample of SecM-stalled ribosomes. We had similar observation (*Figure 1—figure supplement 3*). In our dataset, there is a minor population (13%) of particles that resulted in a low-resolution map (8.6 Å) of an unrotated RNC with two tRNAs, one in the A/A-site and the other in the P/P-site. This structure is likely a snapshot of the SecM-arrested ribosomes with a prolyl-tRNA at the A-site (before peptide-bond formation). However, due to the resolution limitation, the nascent chain density of the P/P-tRNA is relatively weak and thus could not be quantitatively analyzed.

Notably, a very recent study reported high-resolution structures of *Bacillus subtilis* ribosome arrest on MifM peptide (*Sohmen et al., 2015*). Although SecM and MifM show completely no sequence homology, they appear to employ very similar mechanism to inactive the PTC into an uninduced state to cause stalling. This indicates that the stalling mechanism in SecM-Gly-RNC we reported here is highly conserved across species.

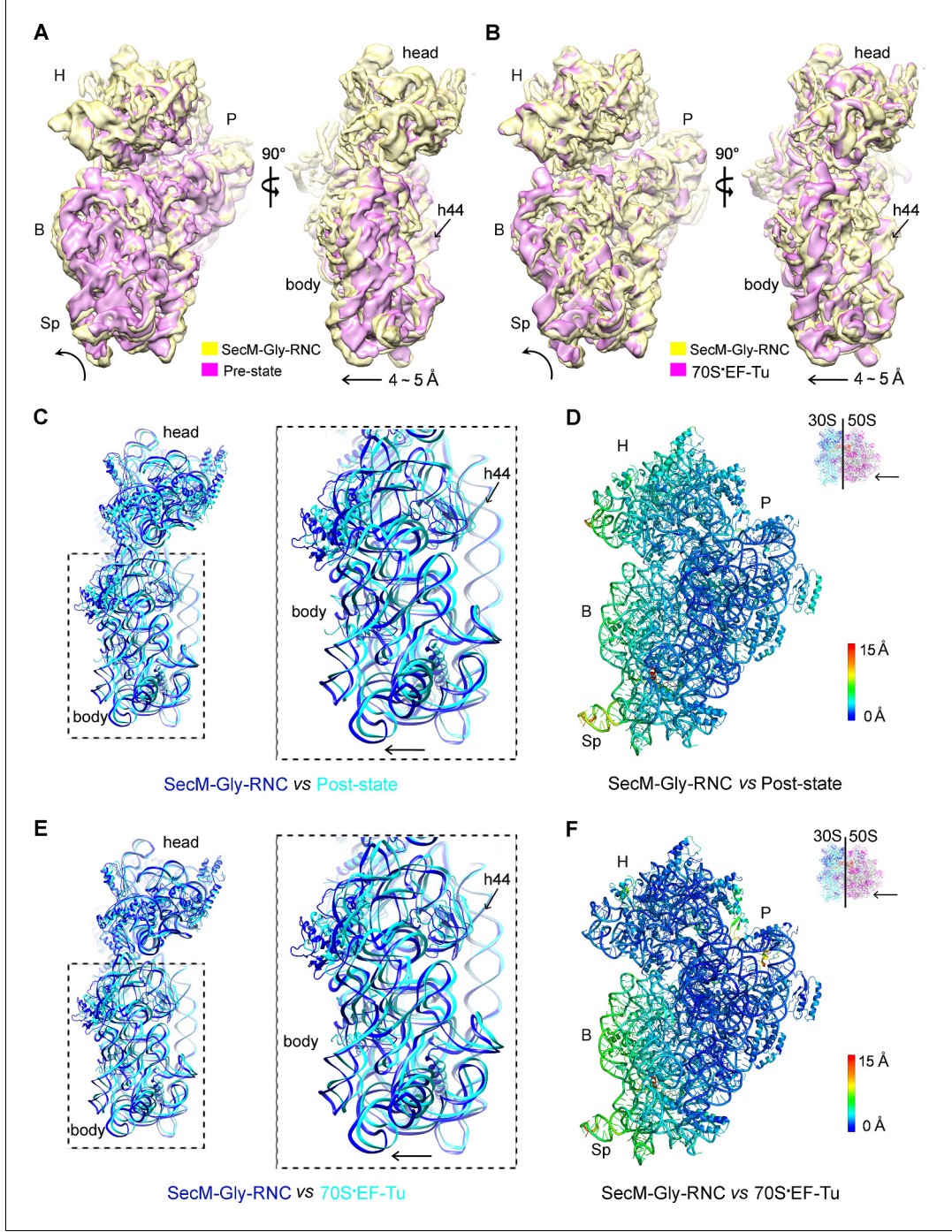

**Figure 6.** A rolling-like motion of the 30S subunit in SecM-Gly-RNC. (A) Comparison of the cryo-EM density map of the 30S subunit in SecM-Gly-RNC (yellow) with that of a pre-translocational state ribosome with three tRNA-bound (EMDB 5796, orchid) (*Brilot et al., 2013*) from an intersubunit view (left panel) and a side view (right panel). (B) Same as (A), but compared with the 30S subunit from the cryo-EM density map of a 70S ribosome bound with elongation factor-Tu (EF-Tu) (EMDB 5036, orchid) (*Villa et al., 2009*). Alignments were done using the segmented 50S subunit as reference. (C) Comparison of the atomic model of the 30S subunit in SecM-Gly-RNC (cyan) with that of the crystal structure of a post-translocational/initiation state ribosome (PDB ID 4V9D, blue) (*Dunkle et al., 2011*). A side view of two superimposed models is shown on the left, with a zoom-in view on the right. (D) Temperature map of the 30S subunit in SecM-Gly-RNC, compared with the 30S subunit in the crystal structure of a post-translocational/initiation state ribosome (PDB ID 4V9D). The 30S subunit in SecM-Gly-RNC is colored according to its distance deviations from 4V9D, with coloring scheme showing on the right. (E) Same as (C), but for

*Figure 6 continued on next page*

*Figure 6 continued*

the comparison between SecM-Gly-RNC (cyan) and the atomic model of the cryo-EM structure of a post-translocational ribosome bound with EF-Tu (PDB ID 4V69, blue) (*Villa et al., 2009*). (**F**) Same as (**D**), but the temperature map is for the comparison in (**E**). The alignment was done using the 23S rRNA residues 1600–2800 as reference. The RMS deviations between respective reference sequences are 1.7 Å for 4V9D vs SecM-Gly-RNC (**C**, **D**), and 1.4 Å for 4V69 vs SecM-Gly-RNC (**E,F**). EF-Tu: Elongation factor-Tu.

The following figure supplements are available for figure 6:

**Figure supplement 1.** The rolling-like motion of the 30S subunit in SecM-Gly-RNC, compared with crystal structures of various ribosomal complexes from *Thermus thermophilus*.

**Figure supplement 2.** Rotation of the 30S subunit relative to the 50S subunit in SecM-Pro-RNC.

**Figure supplement 3.** Map density of decoding centers in SecM-stalled RNCs.

## Mechanism of SecM-induced translation arrest in SecM-Pro-RNC

In the present study, we also determined the structure of SecM-Pro-RNC in rotated conformation at 3.3 Å resolution. In the previous cryo-EM study of SecM-stalled ribosomes, a similar fraction of ribosomes in hybrid state was also observed. But, due to the lack of sufficient resolution, it was not explicitly discussed (*Bhushan et al., 2011*).

Our hypothesis of SecM-Pro-RNC being a stalled form is supported by previous biochemical and recent FRET data. First, using oligonucleotide probes that are highly specific for either tRNA$^{Gly}$, tRNA$^{Pro}$, or tRNA$^{Ala}$, it was found that stalling can occur at either G165 or P166 because there were additional bands detected for arrest sequences (*Muto et al., 2006*), and the fraction of peptidyl-tRNA$^{Pro}$ increased markedly when the SecM construct was truncated to end at P166 (*Muto et al., 2006*). Second, the intersubunit FRET experiments (*Tsai et al., 2014*) show that the lifetime of the rotated-state ribosome significantly increased when the codon of P166 is at the A-site.

Therefore, combining our structural observations in SecM-Pro-RNC, a stalling mechanism in SecM-Pro-RNC could be outlined in the below (*Figure 7B*). Interactions between the nascent chain and the tunnel in SecM-Pro-RNC are considerably different at atomic level from those in SecM-Gly-RNC. At the middle region, W155 and F150 interact with different components of the tunnel (*Figure 3F,G* and *Figure 3—figure supplement 1D*); however, the movements of W155 and F150 are smaller than the length between two Cα atoms of the mainchain, which leads to a further compaction of the C-terminal half of the SecM peptide compared with that in the SecM-Gly-RNC. The more compact conformation of the C-terminal half alters the interaction network between SecM and the ribosome as exemplified by the extensive interactions between R163, I162, Q158, and Q160 with the tunnel (*Figure 4D,E* and *Figure 3—figure supplement 1B,C*). These interactions do not significantly change the conformation of the ribosome, but lead to a distortion on the CCA-end of the A/P*-tRNA and thereby disrupting the canonical base paring between the CCA-end and the ribosomal P-loop. As shown in previous FRET data, during translation of SecM peptide, the lifetime of the rotated state of the ribosome significantly increases between codons, requiring increased EF-G sampling without inhibiting EF-G binding (*Tsai et al., 2014*). Therefore, the impaired accommodation of the A/P-tRNA into the ribosomal P-loop might be responsible for observed delay in tRNA translocation from hybrid A/P site into classical P/P site in the FRET data (*Figure 3B* and *Figure 3—figure supplement 1*). This indicates, in addition to its inhibitive effect on peptide bond formation, SecM also slows down tRNA translocation.

It should be noted that, similar as previous studies of ribosome stalling (*Bhushan et al., 2011*; *Frauenfeld et al., 2011*; *Halic et al., 2006*; *Yap and Bernstein, 2009*), chloramphenicol was added to maximally preserve the RNC during sample preparation. Chloramphenicol is known to inhibit translation by binding near the PTC (*Dunkle et al., 2010*). In our structures, we could observed chloramphenicol density in the map of SecM-Pro-RNC, but not in SecM-Gly-RNC (*Figure 1—figure supplement 7*). Notably, this is in contrast to the crystal structure where chloramphenicol binds to unrotated ribosome (*Dunkle et al., 2010*). Chloramphenicol is observed to interact with C2452/A2503/G2061 of the 23S rRNA in our structure (*Figure 1—figure supplement 7D*), instead of

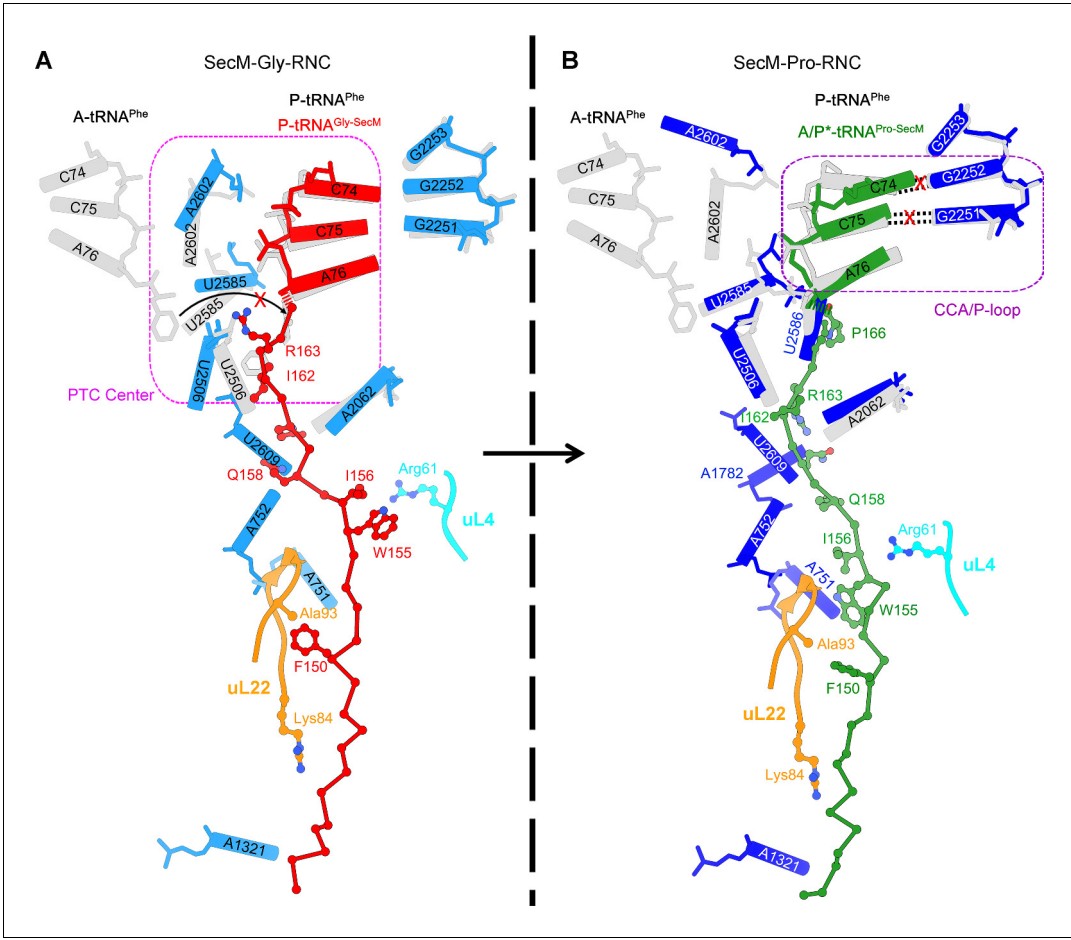

**Figure 7.** Schematic models of SecM-mediated translation stalling. (**A**) Model illustrating how interactions of SecM within the ribosomal tunnel in SecM-Gly-RNC promote conformational rearrangements of the 23S rRNA nucleotides of U2585, U2506, and A2602, and thereby locks the PTC in the inactive state and not optimized for stable binding and accommodation of the incoming A-site tRNA. (**B**) Model illustrating how interactions of SecM within the ribosomal tunnel in SecM-Pro-RNC alter the interaction between the A/P*-tRNA and the P-loop. The altered P-loop interaction increases the life time of A/P*-tRNA and slows down its translocation to classical P/P site.

C2452/G2505/A2062 in the crystal structure (*Dunkle et al., 2010*). Additionally, critical residues of U2585, U2506, and A2602 in our structure show apparently different conformations from that in the crystal structure (*Figure 4—figure supplement 1*). Although we could not accurately estimate how much chloramphenicol has contributed to the ribosome stalling in SecM-Pro-RNC, it appears that it is not the determining factor, given its non-specific effects to the identity of translating peptides.

## Common mechanism of PTC inactivation by regulatory nascent peptides

In this work, our analysis of ribosome–SecM interactions at near-atomic level reveals two different mechanisms of stalling and provides rich details for how SecM nascent peptide could module the translation in distinct steps. Together with previous and recent structural studies of ribosomes stalled through different ways, including antibiotics (for examples, see refs (*Jenner et al., 2013*; *Nguyen et al., 2014*; *Schluenzen et al., 2006*; *Wilson, 2014*)), leader peptides (such as SecM, Erm, TnaC, MifM) (*Arenz et al., 2014a*; *Arenz et al., 2014b*; *Bhushan et al., 2011*; *Bhushan et al., 2010*; *Bischoff et al., 2014*; *Seidelt et al., 2009*; *Sohmen et al., 2015*), it shows that induced conformational changes of the ribosome by nascent peptide and/or small ligand could fine tune translation rate in various mechanisms.

Especially, the mechanism of the PTC inactivation in SecM-Gly-RNC is highly similar to other systems of nascent peptide-mediated stalling, including TnaC (*Bischoff et al., 2014*), MfiM (*Sohmen et al., 2015*), ErmCL (*Arenz et al., 2014a*), and ErmBL (*Arenz et al., 2014b*), indicating that a few PTC residues such as A2062, U2585, U2586, and A2602 are common responsive elements which take conformational signal from nascent chains within the tunnel to modulate the kinetics of translation elongation. This could have a deeper implication in translation control, as it highlights a means of the ribosome to adjust translation rate constantly in codon-specific manners when moving along mRNAs.

## Materials and methods

### Purification of SecM Stalled RNCs

Two similar SecM constructs (*Figure 1—figure supplement 1A*) were generated by the overlapping PCR (polymerase chain reaction). In the first step, the 2xStrep-TEV fragment was amplified using a pair of primers: a forward primer containing 2xStrep-TEV (5'-AAACATATGGCAAGTTGGAGC-3') and a reverse one containing OmpA-TEV (5'-GAGAATCTATACTTCCAAGGTATGAAAAAGACAGC-TATCGCG-3') with the pASG-IBA vector (IBA) as the template. In the second step, the OmpA-Myc-SecM fragment was amplified using a pair of primers: a forward primer containing TEV-OmpA (5'-C-AAGGTATGAAAAAGACAGCTATCGCGATTGCAGTG-3') and a reverse one containing OmpA-Myc-SecM (5'-CTACCGTAGCGCAGGCCGAACAGAAACTGATCTCTGAAGAAGACCTGTTCAGCACGC-CCGTCTGGATAAGCCAGGCGCAAGGCATCCGTGCTGGCCCTCAACGCCTCACCTAACTC-GAGTTT-3') with a previous pET-21b-OmpA construct (*Zhou et al., 2012*) as the template. For the amplification of the OmpA-Myc-SecM-6XHis fragment, the same forward primer and the template as for the OmpA-Myc-SecM fragment were used, while the reverse primer is changed to a primer containing OmpA-Myc-SecM-6XHis (5'- CTACCGTAGCGCAGGCCGAACAGAAACTGATCTCTGAA-GAAGACCTGTTCAGCACGCCCGTCTGGATAAGCCAGGCGCAAGGCATCCGTGCTGGCCCTCAA-CGCCTCACCCTCGAGTTT-3'). In the final step, the two fragments obtained above, 2xStrep-TEV and OmpA-Myc-SecM, were mixed to serve as the template, and a PCR was carried out with two primers 2xStrep-TEV and OmpA-Myc-SecM to amplify the 2xStrep-TEV-OmpA-Myc-SecM fragment. The 2xStrep-TEV-OmpA-Myc-SecM-6XHis fragment was similarly amplified. Both fragments were cloned into pET-21 vectors. Constructs were confirmed by DNA sequencing. SecM-RNCs were generated using an *E.coli* in vitro S30-T7 high-yield protein translation system (Promega Corporation, Madison, WI) according to the manufacturer's instruction.

The RNCs were purified as previously described (*Bhushan et al., 2011*) with modifications. Briefly, a 500-μl reaction was incubated at 30°C for 15 min on a shaker, and then spun through 500 μl sucrose cushion (50 mM HEPES, 250 mM KOAc, 25 mM Mg[OAc]$_2$, 5 mM DTT, 750 mM sucrose, 0.1% Nikkol, 500 μg/ml chloramphenicol, 0.2 U/ml RNasin [Promega Corporation ] and 0.1% pill/ml Complete EDTA-free Protease inhibitor cocktail [Roche, Indianapolis, IN], pH 7.0) at 35,500 g for 60 min in a TLA120.1 rotor (Beckman Coulter, Brea, CA) at 4°C. The pellet was resuspended in 1000 μl of ice-cold 250 buffer (50 mM HEPES, 250 mM KOAc, 25 mM Mg[OAc]$_2$, 5mM DTT, 250 mM sucrose, 0.1% Nikkil, 500 μg/ml chloramphenicol, 0.2 U/ml RNasin, 0.1% pill/ml Complete EDTA-free Protease inhibitor cocktail, pH 7.0) for 60 min at 4 °C, then transferred onto 500 μl of Strep Affinity Resin (IBA) and incubated for 2 hr at 4°C. The resin was then washed with 5 ml of ice-cold 250 buffer and 2 ml 500 buffer (250 buffer with 500 mM KOAc). RNCs were eluted with 1.2 ml of 250 buffer containing 0.5 mM desthiobiotin. Each 600 μl eluted RNCs were spun through 400 μl of a high-salt sucrose cushion at 55,000 r.p.m. for 4 hr in a TLA 55 rotor (Beckman Coulter) at 4°C. The RNC pellet was resuspended in 150 μl of grid buffer (20 mM HEPES, 50 mM KOAc, 6 mM Mg[OAc]$_2$, 1 mM DTT, 500 μg/ml chloramphenicol, 0.05% Nikkol, 0.5% pill/ml Complete EDTA-free Protease inhibitor cocktail, 0.1 U/ml RNasin, and 125 mM sucrose, pH 7.0).

Initial attempts to image the RNCs obtained above by cryo-EM were hindered due to the presence of some polysomes. Therefore, improved sample purification was achieved by adding 20 U of RNaseA (incubated for less than 5 min) into the reaction before the first spinning through sucrose cushion to convert polysomes into monosomes. As a result, polysomes in SecM-stalled RNCs decreased dramatically and ribosomes enriched with strep-affinity resin remained intact as confirmed with cryo-EM. NuPAGE (Life Technologies, Thermo Fisher Scientific , Waltham, MA) and Western

blotting confirmed that the peptidyl-tRNA was a single band. Purified SecM-RNCs were aliquoted in small volumes, flash frozen in liquid nitrogen, and stored at -80°C.

## LC/MS/MS analysis

To analyze the compositions of SecM-stalled RNCs, nascent peptides were purified from SecM-stalled RNCs. The SecM-stalled RNCs were first incubated in grid buffer with 10 mM EDTA and 5% pill/ml Complete EDTA-free Protease inhibitor cocktail to disassemble ribosomes for 30 min at 4°C, then treated with 100 U of RNase A to digest all forms of RNA for another 30 min at 4°C. The mixture was transferred onto Strep Affinity Resin (IBA) and incubated for 1 hr at 4°C. The resin was then washed with ice-cold 250 buffer and 500 buffer (both with 10 mM EDTA). Nascent peptides were eluted with 250 buffer containing 0.5 mM desthiobiotin (with 10 mM EDTA). Purified nascent peptides were analyzed by NuPAGE and Western blotting with primary antibody against Myc (Cell signaling Technology, Danvers, MA) or Strep (Abcam, Cambridge, MA). In-gel digestion of the nascent peptide by lysyl Endopeptidase was carried out following the manufacturer protocol (#125-05061, Wako Pure Chemical Industries, Ltd., Osaka, Japan).

For LC-MS/MS analysis, digestion products were separated by a 60-min gradient elution at a flow rate of 0.300 µL/min with the EASY-nLC 1000 system, which was directly interfaced with a Thermo Orbitrap Fusion mass spectrometer (Thermo Fisher Scientific, Waltham, MA). Mobile phase A consists of 0.1% formic acid and mobile phase B consist of 100% acetonitrile and 0.1% formic acid. The Orbitrap Fusion mass spectrometer was operated in the data-dependent acquisition mode using Xcalibur 3.0 software, and there was a single full-scan mass spectrum in the Orbitrap (350–1550 m/z, 120,000 resolution) followed by top-speed MS/MS scans in the Ion-trap. The MS/MS spectra from each LC-MS/MS run were searched against the selected database using Proteome Discovery searching algorithm (version 1.4).

## EM data acquisition

### Grid preparation and data collection

Freshly purified SecM-RNCs (1.5–2 $OD_{260}$/ml) were diluted by 20-fold, and aliquots of 3.5 µl were applied to glow-discharged Quantifoil R2/2 holey carbon grids (Quantifoil Micro Tools GmbH, Jena, Germany) coated with homemade thin continuous carbon film. The grids were then blotted for 3.5 s and plunged into liquid ethane using an FEI Vitrobot (FEI, Hillsboro, OR) . The grids were imaged with FEI Titan Krios, operated at 300 kV and nominal magnification of 22,500, and equipped with a K2 Summit electron counting camera (Gatan, Pleasanton, CA). The final physical pixel size is 1.32 Å at the specimen level. Defocus values were ranged from -1 to -3.5 µm for data collection. All dose-fractionated cryo-EM images were recorded using UCSF-Image4, a semi-automated low-dose acquisition program (*Li et al., 2013*).

## Image processing

To correct for beam-induced movements, the 14 movie frames (2–15) for each micrograph were aligned using the algorithm developed by Li et al (*Li et al., 2013*). The swarm tool in the e2boxer.py program of EMAN2 (*Tang et al., 2007*) was used for semiautomatic picking of 238,212 particles from 3,908 micrographs. Contrast transfer function parameters were estimated using CTFFIND3 (*Mindell and Grigorieff, 2003*), and all 2D and 3D classifications and refinements were performed using RELION (*Scheres, 2012*). We used reference-free two-dimensional class averaging and three-dimensional classification to discard bad particles and 192,122 particles were selected for further three-dimensional refinement (*Supplementary file 1*).

A previous cryo-EM map of the empty 70S ribosome was low-pass filtered to 70 Å and used as the reference for the following 3D classification. In the initial 3D classification, particles were split into six groups using an angular sampling of 7.5°. Two major classes were separated based on the large conformational difference of the 30S subunit and positions of tRNAs (*Figure 1—figure supplement 3*). Particles from these two major classes were combined and subjected to further 3D classification (five classes) with a final angular sampling of 1.8°. Three major classes were separated and showed improved resolution for the 70S ribosome with clear densities of nascent peptide chains in the exit tunnel. As a result, two of them were combined to yield a subset of 60,354 particles for SecM-Gly-RNC, and the other one yielded 41,501 particles for SecM-Pro-RNC. To improve the

density quality of the nascent peptide chain, a soft mask of the 50S subunit (plus tRNA) was applied during structural refinement, along with an angular sampling of 0.9° combined with local angular searches around the refined orientations (*Figure 1—figure supplement 3*). Soft masks were made by converting atomic models into density maps, and adding cosine-shaped edges.

Reported resolutions were based on the gold standard FSC = 0.143 criterion, and FSC curves (*Figure 1—figure supplement 4*) were corrected for the effects of a soft mask on the FSC curve using high-resolution noise substitution (*Chen et al., 2013*). The resulting density maps were corrected for the modulation transfer function of the detector and sharpened using postprocessing options of RELION 1.3.

## Model building and refinement

Modeling of the structures of SecM-stalled RNCs was based on the crystal structure of the *E.coli* 70S ribosome (PDB ID 4V7T) (*Dunkle et al., 2010*). First, the crystal structure was docked by rigid body fitting into the Cryo-EM density maps of SecM-Gly-RNC and SecM-Pro-RNC using EMfit (*Rossmann et al., 2001*) and UCSF Chimera (*Pettersen et al., 2004*). Nascent chains were manually built using COOT. The overall fitting of the crystal structure showed an excellent agreement with our map densities. Then minor adjustments of the side chains of uL4 and uL22, and nucleotides of 16S rRNA, 23S rRNA, and tRNAs, were manually performed using COOT (*Emsley et al., 2010*). Next, the models were further refined against the density maps with stereochemical and secondary structure restraints using Phenix.real_space_refine (*Adams et al., 2010*). At last, the refined models were subject to REFMAC (*Murshudov et al., 1997*) for further refinement in Fourier space, according to methods previously described (*Amunts et al., 2014*; *Fernandez et al., 2014*). To avoid overfitting, different weights for refinement were tested. For refinement of the atomic models of the 50S subunit (50S + tRNA + mRNA + SecM), the models were refined against the 50S-masked maps. After refinement, models of the 50S half were combined with fitted the 30S half to interpret the 70S maps. Final models were evaluated using MolProbity (*Chen et al., 2010*), and statistics of reconstruction and model refinement were provided in *Supplementary file 1*. For cross-validation against overfitting, we followed the procedure previously described (*Fernandez et al., 2014*). This procedure involved the use of both 'half maps' that were calculated from the same halves of the particles as used for the gold-standard FSC calculations. To remove potential model bias from the final model that was built based on the density map from all particles, the atoms coordinates were displaced randomly by up to a maximum of 0.5 Å using PHENIX. This displaced model was then refined against one of the half maps using REFMAC with secondary structure, base pair and planarity restraints applied. FSC curves were calculated between the resulting model and the map it was refined against ($FSC_{work}$, red curves in *Figure 1—figure supplement 8*), and between the resulting model and the other half map ($FSC_{test}$, blue curves in *Figure 1—figure supplement 8*). The small separation between work and test FSC curves suggested that the models were not overfitted. Structural analysis and figure preparation were done with Pymol or Chimera.

The density maps of SecM-Gly-RNC and SecM-Pro-RNC (with 50S-based mask applied during refinement) have been deposited in the EMData Bank under accession code of 6483 and 6486, respectively, and their associated atomic models have been deposited in the Protein Data Bank under accession code of 3JBU and 3JBV, respectively. The density maps (without 50S-based mask) of SecM-Gly-RNC and SecM-Pro-RNC have also been deposited in the EMData Bank under accession code of 6484 and 6485, respectively.

## Acknowledgements

We thank J-W Wang, H-W Wang and X-M Li for helpful discussion, and X-C Bai and IS Fernández for critical comments on the manuscript. We also thank J-L Lei and Y Xu for the EM support, T Yang and Y Wang for computer support. We acknowledge the Tsinghua University Branch of China National Center for Protein Sciences Beijing and the 'Explorer 100' cluster system of the Tsinghua National Laboratory for Information Science and Technology for providing EM and computation resources. This work was funded by grants from the National Basic Research Program of China (2011CB910500 to S-FS) and the National Natural Science Foundation of China (31230016 to S-FS, 31470722 and 31422016 to NG).

## Additional information

### Funding

| Funder | Grant reference number | Author |
|---|---|---|
| National Basic Research Program of China | 2011CB910500 | Sen-Fang Sui |
| National Natural Science Foundation of China | 31230016 | Sen-Fang Sui |
| National Natural Science Foundation of China | 31470722 | Ning Gao |
| National Natural Science Foundation of China | 31422016 | Ning Gao |

The funders had no role in study design, data collection and interpretation, or the decision to submit the work for publication.

### Author contributions

JZ, designed the experiments, performed the experiments, analyzed the data and prepared the figures and movies, prepared the manuscript; XP, designed the experiments; KY, helped modeling and refinement; SS, NG, S-FS, prepared the manuscript

## Additional files

### Supplementary files

• Supplementary file 1. Statistics of data processing and model refinement.

### Major datasets

The following datasets were generated:

| Author(s) | Year | Dataset title | Dataset URL | Database, license, and accessibility information |
|---|---|---|---|---|
| Jun Zhang, Xijiang Pan, Kaige Yan, Shan Sun, Ning Gao, Sen-Fang Sui | 2015 | Mechanisms of ribosome stalling by SecM at multiple elongation steps. | http://www.rcsb.org/pdb/explore/explore.do?structureId=3jbu | Publicly available at the RCSB Protein Data Bank (accession no. 3jbu) |
| Jun Zhang, Xijiang Pan, Kaige Yan, Shan Sun, Ning Gao, Sen-Fang Sui | 2015 | Mechanisms of ribosome stalling by SecM at multiple elongation steps. | http://www.rcsb.org/pdb/explore/explore.do?structureId=3jbv | Publicly available at the RCSB Protein Data Bank (accession no. 3jbv) |
| Jun Zhang, Xijiang Pan, Kaige Yan, Shan Sun, Ning Gao, Sen-Fang Sui | 2015 | Mechanisms of ribosome stalling by SecM at multiple elongation steps. | http://www.ebi.ac.uk/pdbe/entry/emdb/EMD-6483/ | Publicly available at The Electron Microscopy Data Bank (accession no. EMD-6483) |
| Jun Zhang, Xijiang Pan, Kaige Yan, Shan Sun, Ning Gao, Sen-Fang Sui | 2015 | Mechanisms of ribosome stalling by SecM at multiple elongation steps. | http://www.ebi.ac.uk/pdbe/entry/emdb/EMD-6484/ | Publicly available at The Electron Microscopy Data Bank (accession no. EMD-6484) |
| Jun Zhang, Xijiang Pan, Kaige Yan, Shan Sun, Ning Gao, Sen-Fang Sui | 2015 | Mechanisms of ribosome stalling by SecM at multiple elongation steps. | http://www.ebi.ac.uk/pdbe/entry/emdb/EMD-6485/ | Publicly available at The Electron Microscopy Data Bank (accession no. EMD-6485) |
| Jun Zhang, Xijiang Pan, Kaige Yan, Shan Sun, Ning Gao, Sen-Fang Sui | 2015 | Mechanisms of ribosome stalling by SecM at multiple elongation steps. | http://www.ebi.ac.uk/pdbe/entry/emdb/EMD-6486/ | Publicly available at The Electron Microscopy Data Bank (accession no. EMD-6486) |

The following previously published datasets were used:

| Author(s) | Year | Dataset title | Dataset URL | Database, license, and accessibility information |
|---|---|---|---|---|
| Schmeing TM, Huang KS, Strobel SA, Steitz TA | 2005 | The structure of c-hpmn and CCA-PHE-CAP-BIO bound to the large ribosomal subunit of haloarcula marismortui | http://www.rcsb.org/pdb/explore/explore.do?structureId=1vq6 | Publicly available at the RCSB Protein Data Bank (Accession no: 1vq6) |
| Schmeing TM, Huang KS, Strobel SA, Steitz TA | 2005 | The structure of CC-HPMN AND CCA-PHE-CAP-BIO bound to the large ribosomal subunit of haloarcula marismortui | http://www.rcsb.org/pdb/explore/explore.do?structureId=1vqn | Publicly available at the RCSB Protein Data Bank (Accession no: 1vqn) |
| Selmer M, Dunham CM, Murphy FV, Weixlbaumer A, Petry S, Kelley AC, Weir JR, Ramakrishnan V | 2006 | Structure of the Thermus thermophilus 70S ribosome complexed with mRNA, tRNA and paromomycin | http://www.rcsb.org/pdb/explore/explore.do?structureId=4v51 | Publicly available at the RCSB Protein Data Bank (Accession no: 4v51) |
| Villa E, Sengupta J, Trabuco LG, LeBarron J, Baxter WT, Shaikh TR, Grassucci RA, Nissen P, Ehrenberg M, Schulten K, Frank J | 2009 | Ternary complex-bound E.coli 70S ribosome | http://www.rcsb.org/pdb/explore/explore.do?structureId=4v69 | Publicly available at the RCSB Protein Data Bank (Accession no: 4v69) |
| Gao YG, Selmer M, Dunham CM, Weixlbaumer A, Kelley AC, Ramakrishnan V | 2009 | The structure of the ribosome with elongation factor G trapped in the post-translocational state | http://www.rcsb.org/pdb/explore/explore.do?structureId=4v5f | Publicly available at the RCSB Protein Data Bank (Accession no: 4v5f) |
| Jin H, Kelley AC, Loakes D, Ramakrishnan V | 2010 | Structure of the 70S ribosome bound to Release factor 2 and a substrate analog provides insights into catalysis of peptide release | http://www.rcsb.org/pdb/explore/explore.do?structureId=4v5j | Publicly available at the RCSB Protein Data Bank (Accession no: 4v5j) |
| Voorhees RM, Schmeing TM, Kelley AC, Ramakrishnan V | 2010 | The structure of EF-Tu and aminoacyl-tRNA bound to the 70S ribosome with a GTP analog | http://www.rcsb.org/pdb/explore/explore.do?structureId=4v5l | Publicly available at the RCSB Protein Data Bank (Accession no: 4v5l) |
| Dunkle JA, Xiong L, Mankin AS, Cate JH | 2010 | Crystal structure of the E. coli ribosome bound to chloramphenicol | http://www.rcsb.org/pdb/explore/explore.do?structureId=4v7t | Publicly available at the RCSB Protein Data Bank (Accession no: 4v7t) |
| Jenner LB, Demeshkina N, Yusupova G, Yusupov M | 2010 | Elongation complex of the 70S ribosome with three tRNAs and mRNA | http://www.rcsb.org/pdb/explore/explore.do?structureId=4v6f | Publicly available at the RCSB Protein Data Bank (Accession no: 4v6f) |
| Dunkle JA, Wang L, Feldman MB, Pulk A, Chen VB, Kapral GJ, Noeske J, Richardson JS, Blanchard SC, Cate JH | 2011 | Structures of the bacterial ribosome in classical and hybrid states of tRNA binding | http://www.rcsb.org/pdb/explore/explore.do?structureId=4v9d | Publicly available at the RCSB Protein Data Bank (Accession no: 4v9d) |
| Brilot AF, Korostelev AA, Ermolenko DN, Grigorieff N | 2013 | Structure of the Ribosome with Elongation Factor G Trapped in the Pre-Translocation State (pre-translocation 70S*tRNA structure) | http://www.rcsb.org/pdb/explore/explore.do?structureId=4v7c | Publicly available at the RCSB Protein Data Bank (Accession no: 4v7c) |
| Brilot AF, Korostelev AA, Ermolenko DN, Grigorieff N | 2013 | Structure of the Ribosome with Elongation Factor G Trapped in the Pre-Translocation State (pre-translocation 70S*tRNA*EF-G structure) | http://www.rcsb.org/pdb/explore/explore.do?structureId=4v7d | Publicly available at the RCSB Protein Data Bank (Accession no: 4v7d) |
| Zhou J, Lancaster L, Donohue JP, Noller HF | 2014 | 70S ribosome translocation intermediate containing elongation factor EFG/GDP/fusidic acid, mRNA, and tRNAs trapped in the AP/AP pe/E chimeric hybrid state | http://www.rcsb.org/pdb/explore/explore.do?structureId=4w29 | Publicly available at the RCSB Protein Data Bank (Accession no: 4w29) |

| Polikanov YS, Steitz TA, Innis CA | 2014 | Crystal structure of the Thermus thermophilus 70S ribosome in the post-catalysis state of peptide bond formation containing dipeptydil-tRNA in the A site and deacylated tRNA in the P site | http://www.rcsb.org/pdb/explore/explore.do?structureId=1vy5 | Publicly available at the RCSB Protein Data Bank (Accession no: 1vy5) |
| Villa E, Sengupta J, Trabuco LG, LeBarron J, Baxter WT, Shaikh TR, Grassucci RA, Nissen P, Ehrenberg M, Schulten K, Frank J | 2009 | Aminoacyl-tRNA-EF-Tu-GDP-kir ternary complex-bound E. coli 70S ribosome | http://www.ebi.ac.uk/pdbe/entry/emdb/EMD-5036/ | Publicly available at the Protein Data Bank in Europe (Accession no: EMD-5036) |
| Brilot AF, Korostelev AA, Ermolenko DN | 2013 | Structure of the Ribosome with Elongation Factor G Trapped in the Pre-Translocation State | http://www.ebi.ac.uk/pdbe/entry/emdb/EMD-5796/ | Publicly available at the Protein Data Bank in Europe (Accession no: EMD-5796) |

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
