## [Decision Letter]

Thank you for submitting your work entitled "Mechanisms of Ribosome Stalling by SecM at Multiple Elongation Steps" for peer review at eLife. Your submission has been favorably evaluated by John Kuriyan (Senior Editor) and three reviewers, one of whom, Sjors Scheres, is a member of our Board of Reviewing Editors.

The reviewers have discussed the reviews with one another and the Reviewing editor has drafted this decision to help you prepare a revised submission.

In the present manuscript, the authors describe two cryo-EM reconstructions of SecM-stalled *E. coli* 70S ribosomes, at 3.5-3.9 A resolution. These structures provide the high-resolution views of this prototypical ribosome stalling peptide. They reveal how the stalling mechanism can influence the peptidyl transferase center and exit tunnel in separate steps of the elongation cycle. Given the widespread interest in mechanisms of ribosome pausing and stalling, this manuscript will be of general interest to readers of *eLife*. There are a few points the authors need to address before the paper can be published, as described below.

Essential revisions:

1) One of the main conclusions is that the SecM terminates in Gly in the unrotated state, and in proline in the rotated state, which follows a similar definition in Bhushan et al. However, at this resolution it can be difficult to distinguish between an additional residue and the chain adopting different conformations in each state (especially when the sequence consists of small residues [ala-gly-pro] that have considerable conformational flexibility). The authors use the positions of F150 and W155 to determine the register. However they also note that the movements of these two residues are smaller than the length between two mainchain Cα atoms. Given the importance of distinguishing the peptide termini, clear figures showing the density should be provided for the attachment of these residues to the P-site tRNAs. Can the authors use the tRNA density to discriminate a prolyl-tRNA from a glycyl-tRNA? In addition, it should be shown by mass spectrometry or high-resolution gels that there are two distinct peptides in the EM sample. Currently, Figure 1—figure supplement 1, panels C and D suggest 'only one major band for the peptidyl-tRNA is present', whereas based on the EM classification there should be a 2:3 ratio of proline:glycine. If the assumption that two peptides are present is correct, does the ratio of unrotated state fall with longer incubation times (as proline-tRNA becomes incorporated)?

2) The authors fail to mention the use of chloramphenicol until the Discussion. It should be mentioned in Results, so that readers are aware of its presence.

3) The authors fail to discuss how their observations compare to the MD flexible fitting carried out by Gumbart et al. (2012), which modeled W155 stacking on A751. Were Gumbart et al. modeling into the equivalent of the SecM-Pro-RNC state, or the equivalent of the SecM-Gly-RNC state?

4) In the Discussion, the authors don't compare their results to other stalling peptides in enough depth. For example, there is only a brief mention of the MifM system in Bacillus, and no comparison to the ErmB and ErmC systems. The latter has also been modeled to involve perturbation of the PTC. See PMID: 24662426.

5) The authors provide no description of the methods used for any of the superpositions with other published structures. These methods should be included, along with RMS deviations. This information helps readers understand the significance of any reported distances in observed changes, i.e. in Figure 4.

6) The description of subunit rolling in the subsection “Conformational changes of the ribosome upon SecM recognition by the 50S tunnel” requires more careful comparisons to, for example, Pulk and Cate (PMID: 23812721), where widening between the subunits was noted, and to reconstructions of the eukaryotic ribosome in PMID: 24995983.

7) In Figure 6—figure supplement 3, the 3rd codon base (+6) is modeled syn, it seems. Is this correct? Does this make sense with the complex? Could that pairing with the tRNA explain in part the stalling mechanism in the A/P* state? The authors should comment on this, based on prior structural and biochemical data on codon-anticodon pairings that require syn bases. See for example: PMID: 24995983.

8) The presented FSC curves between the model and the map do not make much sense. The steep fall-offs in the black curves at 0.26 and 0.28 1/Angstrom probably coincide with the highest resolutions included in the refinement, and would thereby indicate that overfitting of the model did take place, despite claims made based on the red (work) and blue (test) curves. The red and blue curves are probably not calculated well. The test curve usually lies below the work curve. In their plot, the authors show the test curve even slightly above the work curve. That would be very surprising. Therefore, refinement of the models and these FSC figures should be re-done.

Minor points:

*Reviewer #2:* 1) The authors conclude that there are no large-scale changes associated with stalling. However, the presence of subunit rolling, which is yet to be shown to be part of the normal translation cycle of prokaryotic ribosomes, suggests that this might be a manifestation of stalling and should not be discounted.

2) There are very few structures of nascent chains trapped within the polypeptide exit tunnel. It would be interesting if the authors could comment on the backbone geometry of the peptide, or at least report the Ramachandran statistics.

3) In the manuscript both ratcheted/unratcheted and rotated/non-rotated are used to describe the state of the ribosome. It might be helpful to use one set of terms consistently.

4) The phrase “might root into the structure of the ribosome itself” (at the end of the subsection “Mechanism of SecM-induced translation stalling in SecM-Gly-RNC”) is not clear.

5) Was the class (15 k particles) containing A and P-site tRNAs also subjected to masking? The presence of this class should be mentioned in the Results section even if the nascent chain density cannot be analysed.

6) Too often X is described to 'interact' with Y. The nature of the interactions between SecM and the ribosome could be more descriptive.

7) Are the maps consistent with the maps from Bhushan et al.? Does their observed ~2 Å shift in the position of the tRNA-NC simply reflect the difficulties in interpreting low-resolution maps or represent a real difference between the two structures?

8) Does altering the length between the bulky residues (F150 and W155) have an impact on stalling efficiency?

9) The map density is occasionally referred to as 'electron density'. In EM electrons are scattered by the Coulomb potential of nuclei rather than by electrons. 'Map density' would be a preferable term.

10) Chloramphenicol should be clearly labeled in Figure 1—figure supplement 7.

11) In Figure 1, it may be helpful to use different colors/shades for the nascent chain and the tRNA.

*Reviewer #3:* In the last paragraph of the subsection “Biochemical sample preparation and cryo-EM structural determination”, the authors should more clearly define what they mean by the SecM-Gly-RNC and SecM-Pro-RNC structures. For those not in the field, it will be confusing. Perhaps citing figure supplements to cartoons at this point would be helpful.

In the same passage, the authors should cite PMID: 25132179, as the Polikanov et al. paper validates the prior Schmeing et al. results which were with 50S subunits, not the intact ribosome. This reference should also be cited the in the second paragraph of the subsection “Mechanism of SecM-induced translation stalling in SecM-Gly-RNC”.

Video 4 and Video 5 should be present as Chimera morphs, rather than toggling. This would make the general motion of the two states compared to the induced state much clearer.

The authors should include at least one micrograph of the raw data as a figure supplement. In addition, it would be helpful to see some maps showing local resolution. The maps are likely better than the quoted resolution in some of the regions of interest, based on separation of nucleotide bases.

Can the authors comment on why they did not use particle polishing, as implemented in Relion? It seems that they could possibly gain some in resolution. Every little bit would help for this kind of atomic modeling.

---

## [Author Response]

*In the present manuscript, the authors describe two cryo-EM reconstructions of SecM-stalled* E. coli *70S ribosomes, at 3.5-3.9 A resolution. These structures provide the high-resolution views of this prototypical ribosome stalling peptide. They reveal how the stalling mechanism can influence the peptidyl transferase center and exit tunnel in separate steps of the elongation cycle. Given the widespread interest in mechanisms of ribosome pausing and stalling, this manuscript will be of general interest to readers of* eLife*. There are a few points the authors need to address before the paper can be published, as described below. Essential revisions:*

*1) One of the main conclusions is that the SecM terminates in Gly in the unrotated state, and in proline in the rotated state, which follows a similar definition in Bhushan et al. However, at this resolution it can be difficult to distinguish between an additional residue and the chain adopting different conformations in each state (especially when the sequence consists of small residues [ala-gly-pro] that have considerable conformational flexibility). The authors use the positions of F150 and W155 to determine the register. However they also note that the movements of these two residues are smaller than the length between two mainchain Cα atoms. Given the importance of distinguishing the peptide termini, clear figures showing the density should be provided for the attachment of these residues to the P-site tRNAs. Can the authors use the tRNA density to discriminate a prolyl-tRNA from a glycyl-tRNA? In addition, it should be shown by mass spectrometry or high-resolution gels that there are two distinct peptides in the EM sample. Currently, Figure 1—figure supplement 1, panels C and D suggest 'only one major band for the peptidyl-tRNA is present', whereas based on the EM classification there should be a 2:3 ratio of proline:glycine.*

To verify the biological composition of the two cryo-EM structures, we performed additional experiments and structural analyses. There are several lines of evidence (listed in the following) supporting our assignment of two structures to SecM-Pro-RNC and SecM-Gly-RNC.

A) To distinguish the peptide termini, we analyzed the local densities of mRNA: tRNA duplex (at the A- and P-site) in our two density maps (Figure 1—figure supplement 5). Although the quality of the map is not enough for ab initio modelling of the nucleotides, it is sufficient to distinguish purines from pyrimidines in many places. The codon of Gly (GGC) contains two purines at positions 1 and 2, while in contrast, the codon of Pro (CCU) is composed of pyrimidines exclusively (Figure 1—figure supplement 5). On the other end, the anti-codon of tRNA_pro_ (GGA) is composed of all purines (Figure 1—figure supplement 5). Our assignment and fitting are highly consistent with the density appearance. If we switch the tRNA identify in our map of SecM-Pro-RNC, an optimal fitting could not be obtained. Especially, the A-site anti-codon of tRNA in SecM-Pro-RNC is best explained by tRNA_pro_, which contains three large bases (Figure 1—figure supplement 5).

B) We also analyzed the local density at the peptidyl-tRNA connections in these two maps (Figure 8). As shown, apparent difference in this region can be observed. For example, a kink in the peptidyl-tRNA connection of SecM-Pro-RNC can be better explained by a proline. Also, as noted in the manuscript, F150 and W155 helped the assignment as well.

Author response image 1.Local density of peptidyl-tRNA connection.**DOI:**
http://dx.doi.org/10.7554/eLife.09684.031

C) We followed the reviewers’ suggestion and employed mass spectrometry and high-resolution electrophoresis to confirm the peptide termini.

Firstly, the high-resolution gel was still not sufficient to tell the difference of Gly-peptide and Pro-peptide. We tried to lower the sample input by 2-fold dilution serials and no separation of bands was observed. This is not unexpected, because they only differ in one amino acid. Nevertheless, it is worth mentioning that Western blotting with two different antibodies (anti-myc and anti-Strep) again confirmed the homogeneity of the sample, indicating that the SecM plasmid construct had indeed arrested the ribosome on its terminal codons.

Author response image 2.(**A**) High-resolution gel and Western blotting analyses of the purified SecM-RNC and (**B**) purified nascent peptide.**DOI:**
http://dx.doi.org/10.7554/eLife.09684.032

Secondly, we subjected the purified nascent peptides to mass spectrometry (Figure 1—figure supplement 2). To purify the nascent peptides, purified RNCs (the same sample used for cryo-EM analysis) were first treated with EDTA and then with excessive RNase A to remove all forms of RNAs. Nascent peptides were enriched and purified with anti-Strep beads (Figure 1—figure supplement 2). Eluted peptides were confirmed with Western blotting (Anti-Myc and Anti-Step). Concentrated (TCA precipitation) sample was then resolved on NuPAGE. Respective band of peptide was recovered from the gel, digested with a lysyl endopeptidase and subjected to a tandem MS/MS analysis.

Lysyl-specific enzyme would generate fragments only containing the C-terminal fragments by cleavage between F150 and K149 (Figure 1—figure supplement 2), which can be detected by MS. Consistent with our conclusion based on structural analysis, there are two species in the sample, with the C-terminus ending at a Gly or a Pro. The relative abundance of two species is ~1: 2 for SecM-Gly: SecM-Pro. Although the ratio of Gly: Pro from mass spectrometry is different from that derived from cryo-EM analysis, the difference can be justified by the following two major reasons: (A) the sample used for mass spectrometry (at the peptide level) has undergone additional purification and processing, and therefore does not represent the same sample space as that used for cryo-EM (at the RNC level); (B) Cryo-EM analysis includes biased particle selection procedures (initial particle picking, 2D classification, 3D classification), and thus does not necessarily reflect the true stoichiometry in the reaction mixture. For example, the assignment of those low-quality particles (with bad 2D averages and low-quality 3D map) to either SecM-Gly-RNC or SecM-Pro-RNC is not possible.

*If the assumption that two peptides are present is correct, does the ratio of unrotated state fall with longer incubation times (as proline-tRNA becomes incorporated)?*

In theory, with longer incubation time, the hybrid A/P-tRNA^pro^ would go into the P/P-site, resulting in unrotated ribosomes. But based on our observation, once tRNA^pro^ assumes a P/P position, the nascent peptide will be released by release factors present in the S30 extract. This is because in our plasmid constructs tandem stop-codons exist following the terminal Proline codon. On the other hand, with increasing incubation time, more proline-tRNA would be incorporated and arrested in hybrid A/P-site, resulting in rotated ribosomes. Therefore, it is hard to predict how the ratio would change, unless some sort of kinetic experiments were performed. Nevertheless, we have carefully chosen the incubation time of 15-min to purify the RNCs based on the maximal fraction of arrested peptidyl-tRNA present in the reaction mixture at this time point (Figure 1—figure supplement 1).

*2) The authors fail to mention the use of chloramphenicol until the Discussion. It should be mentioned in Results, so that readers are aware of its presence.*

We have now mentioned the use of chloramphenicol in the first section of Results.

*3) The authors fail to discuss how their observations compare to the MD flexible fitting carried out by Gumbart et al. (2012), which modeled W155 stacking on A751. Were Gumbart et al. modeling into the equivalent of the SecM-Pro-RNC state, or the equivalent of the SecM-Gly-RNC state?*

The model in Gumbart et al. (2012) was derived by two steps: (A) Fitting of components into the 5.6-Å map of SecM-Gly-RNC (Bhushan et al., 2011) via MDFF, and (B) further refinement and modelling (MDFF) of a subsystem (SecM nascent chain, tRNA, tunnel components) derived from the previous round of MDFF. The major conclusions of Gumbart et al. (2012) are (1) R163 of SecM interacts with the bases of A2062 and U2586; (2) A751 stacks with W155 of SecM; (3) Q158 of SecM interacts with A752, and Q160 with U2609.

Very interestingly, these observed interactions in Gumbart et al., (2012) are highly consistent with our analysis of SecM-Pro-RNC (Figure 2 and related figure supplements), except that we didn’t see direct interaction between R163 and A2062. Therefore, it appears to us that although the starting model of Gumbart et al. is SecM-Gly-RNC (unrotated ribosome), it actually recapitulated our observations from the complex of SecM-Pro-RNC (rotated ribosome). This implies that the interaction pattern we observed in SecM-Pro-RNC is likely a thermodynamically favored state, further supporting our conclusion that SecM-Pro-RNC is another major form of stalled ribosomes by SecM.

We have added relevant discussion in the main text.

*4) In the Discussion, the authors don't compare their results to other stalling peptides in enough depth. For example, there is only a brief mention of the MifM system in Bacillus, and no comparison to the ErmB and ErmC systems. The latter has also been modeled to involve perturbation of the PTC. See PMID: 24662426.*

We added a paragraph discussing the commonality of these ribosome arresting mechanisms (Discussion). In short, although the specific nascent chain-ribosome interactions are different in these cases, a few 23S rRNA residues, such as, U2584-U2586, appear to be the major responsive elements at the PTC.

*5) The authors provide no description of the methods used for any of the superpositions with other published structures. These methods should be included, along with RMS deviations. This information helps readers understand the significance of any reported distances in observed changes, i.e. in Figure 4.*

We used Chimera or Pymol to perform alignments. Reference sequences of the 23S rRNA and RMS deviations were summarized in Table 1. Note that RMS refers to deviation of aligned reference sequences. Methods and reported RMS were now provided in related figure legends when necessary.

Author response table 1.**DOI:**
http://dx.doi.org/10.7554/eLife.09684.033Model1Model2Reference ResiduesRMS(Å)FigureSecM-Gly-RNC4V6FPyMOLalign with residues 1600-2700(23S rRNA)1.902Figure1-supplemental_figure6,Figure6-supplemental_figure1SecM-Gly-RNC4V6FPyMOLalign with residues 10-550(16S rRNA)1.537Figure1-supplemental_figure6SecM-Pro-RNC4V6FPyMOLalign with residues 1600-2900(23S rRNA)1.588Figure1-supplemental_figure6,Figure6-supplemental_figure2SecM-Pro-RNC4V6FPyMOLalign with residues 10-550(16S rRNA)1.330Figure1-supplemental_figure6SecM-Pro-RNC4V6FPyMOLalign with residues 920-1400(16S rRNA)1.198Figure1-supplemental_figure64V9DSecM-Pro-RNCPyMOLalign with residues 10-550(16S rRNA)1.193Figure1-supplemental_figure64W29SecM-Pro-RNCPyMOLalign with residues 10-550(16S rRNA)1.425Figure1-supplemental_figure64V7CSecM-Pro-RNCPyMOLalign with residues 920-1400(16S rRNA)1.196Figure1-supplemental_figure64V7DSecM-Pro-RNCPyMOLalign with residues 920-1400(16S rRNA)1.066Figure1-supplemental_figure6SecM-Gly-RNC4V7TPyMOLalign with residues 1600-2700(23S rRNA)1.535Figure1-supplemental_figure7SecM-Pro-RNC4V7TPyMOLalign with residues 1600-2700(23S rRNA)1.112Figure1-supplemental_figure71VQ61VQNPyMOLalign with residues 2400-2800(23S rRNA)0.308Figure4,Figure4-supplemental_figure1,Figure5-supplemental_figure1SecM-Gly-RNC1VQNPyMOLalign with residues 2400-2800(23S rRNA)1.527Figure4,Figure4-supplemental_figure14V5JSecM-Gly-RNCPyMOLalign with residues 2400-2800(23S rRNA)1.007Figure44V7T1VQNPyMOLalign with residues 2400-2800(23S rRNA)1.054Figure4-supplemental_figure1SecM-Pro-RNC1VQNPyMOLalign with residues 2400-2800(23S rRNA)1.405Figure5,Figure4-supplemental_figure1,Figure5-supplemental_figure15796(EMDB)SecM-Gly-RNCChimerafit with 50S subunit0.9494(correlation)Figure65036(EMDB)SecM-Gly-RNCChimerafit with 50S subunit0.9634(correlation)Figure64V9DSecM-Gly-RNCPyMOLalign with residues 1600-2700(23S rRNA)1.697Figure64V69SecM-Gly-RNCPyMOLalign with 23S rRNA1.344Figure64V51SecM-Gly-RNCPyMOLalign with residues 1600-2900(23S rRNA)1.616Figure6-supplemental_figure14V5FSecM-Gly-RNCPyMOLalign with residues 1600-2900(23S rRNA)1.624Figure6-supplemental_figure14V5LSecM-Gly-RNCPyMOLalign with residues 1600-2900(23S rRNA)1.758Figure6-supplemental_figure1SecM-Gly-RNCSecM-Pro-RNCPyMOLalign with 23S rRNA0.996Figure6-supplemental_figure21VY5SecM-Pro-RNCPyMOLalign with residues 1600-2900(23S rRNA)1.487Figure6-supplemental_figure24V9DSecM-Pro-RNCPyMOLalign with 23S rRNA1.386Figure6-supplemental_figure2

*6) The description of subunit rolling in the subsection “Conformational changes of the ribosome upon SecM recognition by the 50S tunnel” requires more careful comparisons to, for example, Pulk and Cate (PMID: 23812721), where widening between the subunits was noted, and to reconstructions of the eukaryotic ribosome in PMID: 24995983.*

We thank the reviewers for pointing out the previous observation of 30S subunit rolling (Pulk and Cate, 2013). It indeed makes interesting comparison to our structure of SecM-Gly-RNC. One of the crystal structures reported in Pulk and Cate (2013) is an unrotated ribosome bound with EF-G/GMPPCP (PDB ID: 4KJ9/4KJA). Very interestingly, our structure of SecM-Gly-RNC is highly similar to this structure (Figure 10), although the intersubunit space in SecM-Gly-RNC is slightly wider. This observation again suggests that the rolling of the 30S body is indeed a mode of conformational change for the 30S subunit.

Author response image 3.Comparison of SecM-Gly-RNC with an unrotated 70S-EF-G-GMPPCP structure.**DOI:**
http://dx.doi.org/10.7554/eLife.09684.034

We have included this comparison in the main text.

*7) In Figure 6—figure supplement 3, the 3rd codon base (+6) is modeled syn, it seems. Is this correct? Does this make sense with the complex? Could that pairing with the tRNA explain in part the stalling mechanism in the A/P* state? The authors should comment on this, based on prior structural and biochemical data on codon-anticodon pairings that require syn bases. See for example: PMID: 24995983.*

We rechecked the modelling and corrected the configuration of the (+6) position. The initial modelling of mRNA was done with poly-adenine manually in COOT, and replaced with respective nucleotides after the structural analysis. The syn configuration of the (+6) position was somehow introduced unintentionally during the modelling.

*8) The presented FSC curves between the model and the map do not make much sense. The steep fall-offs in the black curves at 0.26 and 0.28 1/Angstrom probably coincide with the highest resolutions included in the refinement, and would thereby indicate that overfitting of the model did take place, despite claims made based on the red (work) and blue (test) curves. The red and blue curves are probably not calculated well. The test curve usually lies below the work curve. In their plot, the authors show the test curve even slightly above the work curve. That would be very surprising. Therefore, refinement of the models and these FSC figures should be re-done.*

We followed the reviewers’ suggestion and re-performed model refinement and validation. We followed a published protocol (Fernandez et al., 2014) to refine the model in Fourier space using REFMAC. At first, the weighting of the experimental map was optimized (Figure 11). Final refinement of the two models was done with these selected values of weighting (Dashed box). The final results were presented in Figure 1—figure supplement 8.

Please refer to the revised manuscript for details in model refinement and validation.

Also, it needs to be mentioned that after model refinement, we did not actually observe large modelling differences from the previous models derived from real-space refinement by PHENIX. As a result, our previous analysis of detailed interactions between SecM and the tunnel still hold true (Figure 2–Figure 5).

Author response image 4.Model Refinement and validation.**DOI:**
http://dx.doi.org/10.7554/eLife.09684.035

*Minor points:* Reviewer #2:

*1) The authors conclude that there are no large-scale changes associated with stalling. However, the presence of subunit rolling, which is yet to be shown to be part of the normal translation cycle of prokaryotic ribosomes, suggests that this might be a manifestation of stalling and should not be discounted.*

We have rephrased the Discussion to avoid confusion. In fact, we meant to state that there is no large-scale changes on the 50S subunit. This is in contrast to a previous low-resolution cryo-EM structure (Mitra et al., 2006). As to the 30S body rolling, we believe that this is likely *not* related to SecM-induced stalling. We had some preliminary structural results (5-6-Å resolution) on the factor-free Pre (three tRNA) and Post (P/P and E/E tRNAs) ribosomes. Comparison of them indicated that the intersubunit spacing of the Post ribosome is larger. It appeared to us that the 30S body rolling in prokaryotic ribosomes is another mode of motions during the normal translation cycle (highly similar to eukaryotic ribosomes as discovered in Budkevich et al., 2014). And this rolling motion is likely controlled by the A-site occupancy of A/A-site tRNA and/or factors.

*2) There are very few structures of nascent chains trapped within the polypeptide exit tunnel. It would be interesting if the authors could comment on the backbone geometry of the peptide, or at least report the Ramachandran statistics.*

We evaluated the models of nascent chains in our structures using MolProbity. The Ramachandran statistics are shown below (Figure 12). In fact, we did not find any residues in disallowed regions. This indicates that the interactions between nascent chains and the tunnel wall components did not alter the normal backbone geometry of the SecM peptide.

Author response image 5.Ramachandra diagrams of the two nascent chains.**DOI:**
http://dx.doi.org/10.7554/eLife.09684.036

*3) In the manuscript both ratcheted/unratcheted and rotated/non-rotated are used to describe the state of the ribosome. It might be helpful to use one set of terms consistently.*

We are sorry for the mixed use of both terms. We have now used unrotated/rotated terms throughout the manuscript.

*4) The phrase “might root into the structure of the ribosome itself” (at the end of the subsection “Mechanism of SecM-induced translation stalling in SecM-Gly-RNC”) is not clear.*

We have deleted this general but vague statement.

*5) Was the class (15 k particles) containing A and P-site tRNAs also subjected to masking? The presence of this class should be mentioned in the Results section even if the nascent chain density cannot be analysed.*

This class (15 k particles) containing A and P-site tRNA was not subjected to 50S mask-based refinement. After a normal refinement, we found that the nascent chain density is relatively weak and highly fragmented. Therefore, these particles were not further analyzed. We have followed the suggestion and discussed the presence of this class in the first section of Results.

*6) Too often X is described to 'interact' with Y. The nature of the interactions between SecM and the ribosome could be more descriptive.*

Thanks for the suggestion. We have changed the wording throughout the manuscript.

*7) Are the maps consistent with the maps from Bhushan et al.? Does their observed ~2 Å shift in the position of the tRNA-NC simply reflect the difficulties in interpreting low-resolution maps or represent a real difference between the two structures?*

There is indeed a different between the two maps (Figure 13). Our map was filtered to 5.6-Å resolution, and aligned with the Bhushan map. As shown, superimposition of our model (red) with the Bhushan map indicates there is a shift of the peptide-tRNA linkage in their map. Therefore, it appears to us that the interpretation in Bhushan et al. is consistent with their data. However, the 2-Å shift is beyond the nominal resolution of their density map. Furthermore, the constructs used to generate RNC were different in the two cases. We had a short one (ending at P166 to synchronize the stalling), while they had a slight longer one (ending at 170). Therefore, in principle, their particles might contain ribosomes stalled after terminal P166.

Author response image 6.Comparison of our map of SecM-Gly-RNC (grey) with the Bhushan map (yellow).The atomic model of our SecM-Gly-RNC is shown in red.**DOI:**
http://dx.doi.org/10.7554/eLife.09684.037

*8) Does altering the length between the bulky residues (F150 and W155) have an impact on stalling efficiency?*

Yes, the length between the bulky residues F150 and W155 is crucial for the stalling efficiency. This was demonstrated previously (Nakatogawa and Ito, 2002).

Author response image 7.Insertion between F150 and S151 or deletion of S151 impairs stalling efficiency (adapted from Nakatogawa and Ito, 2002).**DOI:**
http://dx.doi.org/10.7554/eLife.09684.038

As shown, an alanine was inserted between F150 and S151 (lanes 3-4), which would increase the length between F150 and W155. This insertion indeed decreased the stalling efficiency (compare lane 3 with lane 1). Also, internal deletion of S151 (shortening of the spacing between F150 and W155 by one residue) had similar effect (compare lane 5 with lane 1). Indeed, one of the major conclusions in this study (Nakatogawa and Ito, 2002) is that the exact spacing between relevant residues within SecM is required for efficient stalling.

*9) The map density is occasionally referred to as 'electron density'. In EM electrons are scattered by the Coulomb potential of nuclei rather than by electrons. 'Map density' would be a preferable term.*

We have changed the terms to “map density”.

*10) Chloramphenicol should be clearly labeled in Figure 1—figure supplement 5.*

The figure has been revised.

*11) In Figure 1, it may be helpful to use different colors/shades for the nascent chain and the tRNA.*

We have changed the coloring scheme in Figure 1

Reviewer #3:

*In the last paragraph of the subsection “Biochemical sample preparation and cryo-EM structural determination”, the authors should more clearly define what they mean by the SecM-Gly-RNC and SecM-Pro-RNC structures. For those not in the field, it will be confusing. Perhaps citing figure supplements to cartoons at this point would be helpful.*

We have added necessary introduction of these terms in the Results. Also, a cartoon was added in Figure 1—figure supplement 3.

*In the same passage, the authors should cite PMID: 25132179, as the Polikanov et al. paper validates the prior Schmeing et al. results which were with 50S subunits, not the intact ribosome. This reference should also be cited the in the second paragraph of the subsection “Mechanism of SecM-induced translation stalling in SecM-Gly-RNC”*.

We have corrected the citation in the revision.

*Video 4 and Video 5 should be present as Chimera morphs, rather than toggling. This would make the general motion of the two states compared to the induced state much clearer.*

We followed the suggestion and used “morphs”.

*The authors should include at least one micrograph of the raw data as a figure supplement. In addition, it would be helpful to see some maps showing local resolution. The maps are likely better than the quoted resolution in some of the regions of interest, based on separation of nucleotide bases.*

We have now included one raw micrograph in Figure 1—figure supplement 3. Also, the local resolution maps were provided in Figure 1—figure supplement 4.

*Can the authors comment on why they did not use particle polishing, as implemented in Relion? It seems that they could possibly gain some in resolution. Every little bit would help for this kind of atomic modeling.*

In fact, we have previously tried to use particle polishing. However, despite the extensive computation efforts, the outcomes were not good as one would have expected. Maps reconstructed from polished particles are slightly worse in terms of resolution, although no noticeable density difference were seen. This was also true in a few other ribosome-related projects in our lab. We suspect that the performance of particle polishing might relate to specific parameters used for data collection, e.g. the performance of the microscope/camera. In our case, the dose rate used for data collection on K2 is ~ 8.2 counts per second per Å^2^, and the exposure time is 8 seconds for a total of 32 frames. On the frame level, the dose is ~ 1.5 electrons per Å^2^. We noticed that in two papers by Scheres and Ramakrishnan (Amunts et al., 2014; Brown et al., 2014) particle polishing was used for ribosomal particles and improvement of resolutions were reported. Their data collection was done with Falcon II camera (17 frames, total dose 25 electrons per Å^2^), which appears to be in a comparable dose at the frame level. But, it was not mentioned in their papers that how much particle polishing has improved the resolution. It should be noted that we used same parameters as in their papers (as well as a few other combinations) for particle polishing. At present, we are not sure about the reason why particle polishing did not work as expected in our cases.

Nevertheless, we further improved the resolutions of our maps by optimizing the masks used for 50S-mask based refinement. The previous round was done with default parameters in RELION. Now we were able to push the resolutions to 3.6 and 3.3 Å for SecM-Gly-RNC and SecM-Pro-RNC, respectively.